# The impact of genetic adaptation on chimpanzee subspecies differentiation

**Joshua M. Schmidt**[1,2]*, **Marc de Manuel**[3], **Tomas Marques-Bonet**[3,4,5], **Sergi Castellano**[2,6,7], **Aida M. Andrés**[1,2]*

**1** UCL Genetics Institute, Department of Genetics, Evolution and Environment, University College London, London, United Kingdom, **2** Max Planck Institute for Evolutionary Anthropology, Department of Evolutionary Genetics, Leipzig, Germany, **3** Institut de Biologia Evolutiva (Consejo Superior de Investigaciones Científicas–Universitat Pompeu Fabra), Barcelona, Spain, **4** National Centre for Genomic Analysis–Centre for Genomic Regulation, Barcelona Institute of Science and Technology, Barcelona, Spain, **5** Institució Catalana de Recerca i Estudis Avançats (ICREA), Barcelona, Spain, **6** Genetics and Genomic Medicine Programme, Great Ormond Street Institute of Child Health, University College London (UCL), London, United Kingdom, **7** UCL Genomics, London, United Kingdom

* joshmschmidt1@gmail.com (JMS); a.andres@ucl.ac.uk (AMA)

**Data Availability Statement:** Data generated in the course of this investigation and relevant for the interpretation of the results presented here have been deposited with dryad: Schmidt, Joshua et al.

## Abstract

Chimpanzees, humans' closest relatives, are in danger of extinction. Aside from direct human impacts such as hunting and habitat destruction, a key threat is transmissible disease. As humans continue to encroach upon their habitats, which shrink in size and grow in density, the risk of inter-population and cross-species viral transmission increases, a point dramatically made in the reverse with the global HIV/AIDS pandemic. Inhabiting central Africa, the four subspecies of chimpanzees differ in demographic history and geographical range, and are likely differentially adapted to their particular local environments. To quantitatively explore genetic adaptation, we investigated the genic enrichment for SNPs highly differentiated between chimpanzee subspecies. Previous analyses of such patterns in human populations exhibited limited evidence of adaptation. In contrast, chimpanzees show evidence of recent positive selection, with differences among subspecies. Specifically, we observe strong evidence of recent selection in eastern chimpanzees, with highly differentiated SNPs being uniquely enriched in genic sites in a way that is expected under recent adaptation but not under neutral evolution or background selection. These sites are enriched for genes involved in immune responses to pathogens, and for genes inferred to differentiate the immune response to infection by simian immunodeficiency virus (SIV) in natural vs. non-natural host species. Conversely, central chimpanzees exhibit an enrichment of signatures of positive selection only at cytokine receptors, due to selective sweeps in *CCR3*, *CCR9* and *CXCR6* –paralogs of *CCR5* and *CXCR4*, the two major receptors utilized by HIV to enter human cells. Thus, our results suggest that positive selection has contributed to the genetic and phenotypic differentiation of chimpanzee subspecies, and that viruses likely play a predominate role in this differentiation, with SIV being a likely selective agent. Interestingly, our results suggest that SIV has elicited distinctive adaptive responses in these two chimpanzee subspecies.

(2019), The impact of genetic adaptation on chimpanzee subspecies differentiation, v2, Dryad, Dataset, https://doi.org/10.5061/dryad.zcrjdfn6m.

**Funding:** JMS, SC and AMA were supported by the Max Planck Society (https://www.mpg.de/en). AMA and JMS are supported by a Wellcome Trust ISSF grant (204841/Z/16/Z) https://wellcome.ac.uk/what-we-do/our-work/institutional-strategic-support-fund. SC is funded by NIHR HS&DR Programme (14/21/45) (https://www.nihr.ac.uk/funding-and-support/funding-for-research-studies/funding-programmes/health-services-and-delivery-research/). This work is (partly) funded by the NIHR GOSH BRC (SC). The views expressed are those of the authors and not necessarily those of the NHS, the NIHR or the Department of Health (https://www.gosh.nhs.uk/our-research/nihr-great-ormond-street-hospital-brc). Neither the Max Planck Society, the Wellcome Trust nor the NIHR had any role in the study design, data collection and analysis, decision to publish, or preparation of the manuscript.

**Competing interests:** The authors have declared that no competing interests exist.

## Author summary

Viruses are a major factor driving recent and ongoing natural selection in mammalian genomes. Studying the effects such selection has had on chimpanzee genomes can provide valuable insights into how pathogens are affecting an endangered primate species. As there are many notable examples of cross-species transmission between other primates and humans—including the HIV/AIDS pandemic—these studies can also illuminate mechanisms of adaption to pathogens of medical and economic importance. By investigating patterns of genetic differentiation amongst the four chimpanzee subspecies, we show that genetic differences among some subspecies are due to recent genetic adaptation. The genetic variants selected uniquely in eastern chimpanzees fall disproportionally within genes that differentiate the immune response to infection by simian immunodeficiency virus (SIV) in natural vs. non-natural host species. Thus, we infer that SIV has likely elicited adaptive responses in chimpanzees, building upon a growing body of work suggesting that SIV elicits ongoing selection in African primates.

## Introduction

Chimpanzees (*Pan troglodytes*) are, alongside bonobos, human's closest living relatives–the *Pan* and *Homo* lineages having diverged ~6Myr ago [1]. With a per nucleotide divergence of only ~1% [2], *Pan* and *Homo* also share many aspects of their physiology and behaviour, including susceptibility to some pathogens. Studying chimpanzees can teach us about our species by putting recent human evolution in its evolutionary context i.e. the mode and tempo of adaptation and the pressures driving it.

Selection imposed by pathogens has greatly shaped the long-term history of genetic adaptation in the great apes, including chimpanzees and humans [3, 4]. The interest in recent human evolution [5–9] means that we now also have good catalogues of the main targets of local adaptation in many non-African human populations–albeit one biased towards hard selective sweeps. Earlier analyses of genome-wide patterns of diversity in genic and non-genic sites suggested that adaptation via hard selective sweeps has had a limited direct role in shaping human genomes [10,11], with little evidence that local adaptation has substantially affected human population differentiation [10]–unless inferences are boosted with ancient DNA [11]. For instance, Hernandez et al (2011) found that complete selective sweeps involving non-synonymous substitutions have been rare. However, if classical tests of neutrality are underpowered, they may lead us to under estimate the prevalence of natural selection. In fact, sweeps at non-synonymous sites are still important if background selection is controlled for [12]. More recent advances in both datasets and methods have indicated that we had previously lacked power to reliably identify selected loci, particularly in the case of soft sweeps, which are missed by most classical neutrality tests. For instance, machine learning approaches [13–15] with higher power to identify different types of selective sweeps provide some evidence for more pervasive action of selection in the human genome, in particular soft sweeps, than previously thought [16]. Nonetheless, the relative contributions of drift and selection on human population differentiation, particularly soft sweeps [17], are still a matter of debate and are still to be fully determined.

The focus on humans of genomic studies may also bias our general view of the influence of genetic adaptations in natural populations of primates. We know that genome-wide evidence of positive selection scales positively with effective population size ($N_e$) across great ape genera

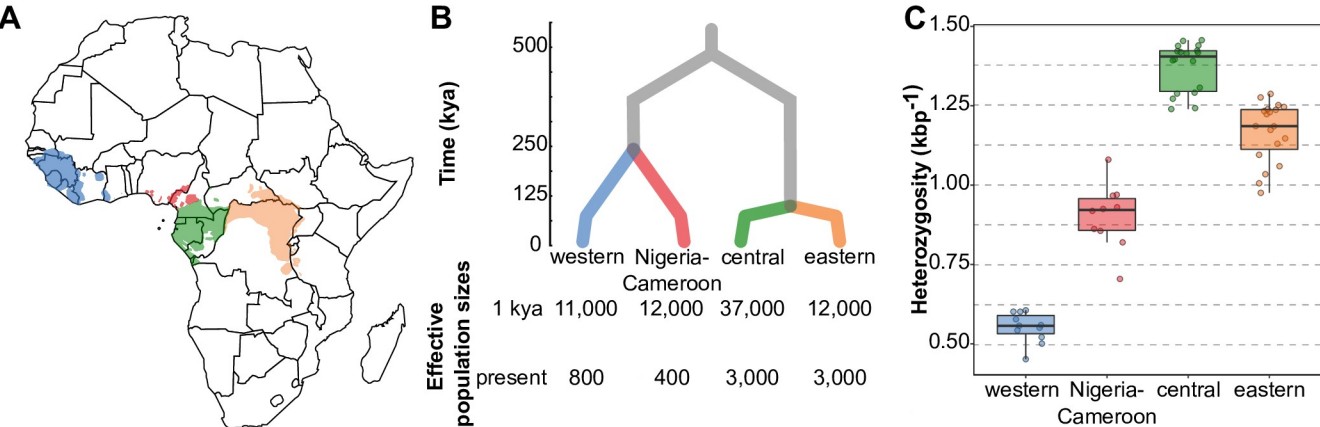

**Fig 1. The geographic distribution and population history of chimpanzees.** A, The ranges of each chimpanzee subspecies within western and central Africa. Range data extracted from the map of chimpanzee geographic range from [21]. Map of Africa modified from public domain source [88]. B, Phylogenetic relationships amongst chimpanzees and the timing of their population divergence, modified from [20]. 1 kya: Long term effective population sizes until 1 kya; present: effective population sizes from 1 kya to present. C, Heterozygosity, reflective of relative differences in effective population sizes. Box plots show median central interquartile range, whiskers the upper and lower interquartile range. Points show individual heterozygosity. For all panels, colour designates subspecies: Blue = western, red = Nigeria-Cameroon, green = central, orange = eastern.

[1, 3, 18, 19], but we largely ignore whether positive selection plays a similarly limited role in shaping other primate genomes as it does in humans. We aim to add to this burgeoning field of knowledge by exploring the recent adaptive history of chimpanzees, by focussing particularly on differentiation caused by divergent selection across the subspecies. This approach does not necessarily capture all the recent adaptations (it is likely insensitive to most potential instances of convergent adaptation), but it provides an excellent global view of the adaptive differences among subspecies, and their selective drivers.

There are four recognised subspecies of chimpanzees, with common names reflecting their location in western and central Africa: eastern, central, Nigeria-Cameroon and western (Fig 1, adapted from [20]). Each chimpanzee subspecies is currently endangered, with western chimpanzees critically so [21]. Subspecies are clearly differentiated, with divergence times ranging from 450 kya to 100 kya, and estimated long-term $N_e$ from 8,000 to 30,000 reflected in varying levels of genetic diversity (Fig 1, adapted from [20]). There is a wide range of ecological variation across the chimpanzee range, which spans over 5,000 km in western and central Africa and includes deep forest and savanna-woodland mosaics. Pathogen incidence can also vary between these groups, as seen recently with the lethal outbreaks of Anthrax [22] and Ebola [23], or the Simian immunodeficiency virus (SIV). SIV, the precursor of the human immunodeficiency virus type 1 (HIV-1) virus that is responsible for the human AIDS pandemic [24], is thought to be largely non-lethal to chimpanzees, although some eastern chimpanzees can develop immunodeficiency, see [25, 26]). Its prevalence is not uniform across the subspecies, and there is no evidence for infections in western or Nigeria-Cameroon chimpanzees [27], while infections have been detected in multiple communities of central and eastern chimpanzees [27, 28]. Given the separate history and differential habitat of each subspecies, and the fact that each subspecies is an independent conservation unit, it is crucial that we identify not only the genetic adaptations shared by all chimpanzees [3], but also the genetic differences conferring differential adaptation to each subspecies.

To do this, we investigated the signatures of recent genetic adaptation in the genomes of the four subspecies by comparing the levels of between subspecies genetic differentiation at genic versus non-genic genomic regions. Such approaches have been utilised before, primarily in the

study of human populations [10, 11, 29], and are not strongly restricted to particular modes of selection. However, we note that certain conservative assumptions are made: such approaches ignore selection acting on functional non-genic sites e.g. enhancers, and that selection on genic sites can affect linked non-genic sites. These conservative assumptions may decrease our estimates of the extent of positive selection, but we are still able to make quantitative comparisons about selection between different subspecies. Given the wide range of $N_e$ between chimpanzees, we also note that selection signals can be obscured in lower $N_e$ subspecies, as random genetic drift influences genetic differentiation; this must be considered when comparing subspecies, and we discuss it with regard to all our inferences (see results). Lastly, differences in average recombination rates between genic and non-genic sites could affect the levels of population differentiation between the two classes of sites, making this an additional factor to control for in selection inferences.

We show that only eastern chimpanzees have an unequivocal genome-wide signal of recent, local positive selection, particularly when compared to central chimpanzees. This adaptation is potentially due to selection on immunity related genes, with evidence consistent with selection imposed by viruses, SIV in particular. In contrast, putative adaptation to SIV in central chimpanzees could be mediated by adaptation in a suite of cell-entry receptors, suggesting the possibility of divergent paths of adaptation to a common pathogen.

## Results

### Genic enrichment in the distribution of derived allele frequency differences

To investigate the influence of recent genetic adaptation in chimpanzee subspecies we compared population differentiation at putatively functional sites (genic sites, defined as +- 2kb from protein-coding genes) to differentiation at non-functional sites (here non-genic). Natural selection can only act on functional sites (although it can affect neutral sites tightly linked to functional sites), so differences between functional and non-functional genomic sites can be ascribed to natural selection. After binning every SNP by its signed difference in derived allele frequency between a pair of subspecies ($\delta$), for each bin of $\delta$ we calculated the genic enrichment, defined as the ratio of genic SNPs vs. all SNPs for each bin of $\delta$, normalized by the global genic SNP ratio [10, 11, 29]. This strategy has been deployed in the study of human local adaptation [10, 11, 29], and by not relying on the patterns of linked variation it is not strongly restricted to particular modes of selection. In all subspecies comparisons (Fig 2, orange line) the genic enrichment is greatest for SNPs with the largest $\delta$, and the tail bins of $\delta$ exhibit significantly greater genic enrichments than any other bin. While not every genic SNP is in this bin due to positive selection, we expect these SNPs, which show the largest frequency differences between subspecies in the genome, to be strongly enriched in targets of positive selection that rose quickly in frequency in one of the two subspecies [10, 11, 29].

The number of tail SNPs and the magnitude of genic enrichment in the tails of $\delta$ across subspecies pairs varies in accordance with their $N_e$ and divergence times (Fig 2 and S1 Table). Calculated against western chimpanzees, the subspecies with the lowest long-term $N_e$ [1, 20], the $\delta$ tail genic enrichment is the least, ranging from 1.05 to 1.10 (Fig 2A). This indicates that this strategy to identify the targets of local adaptation has reduced power in this subspecies. A greater tail genic enrichment, 1.21 to 1.29, is seen for $\delta$ calculated using Nigeria-Cameroon, the species with the second lowest long-term $N_e$ (Fig 2A). This is comparable to the magnitude of the genic enrichment in the tails of $\delta$ between human populations (S1 Appendix, S1 Fig; see [10, 11, 29]; the genic enrichment across each bin of $\delta$ also resembles those observed in human populations (S1 Appendix, S1 Fig; see [10, 11, 29]. In these comparisons the tail genic

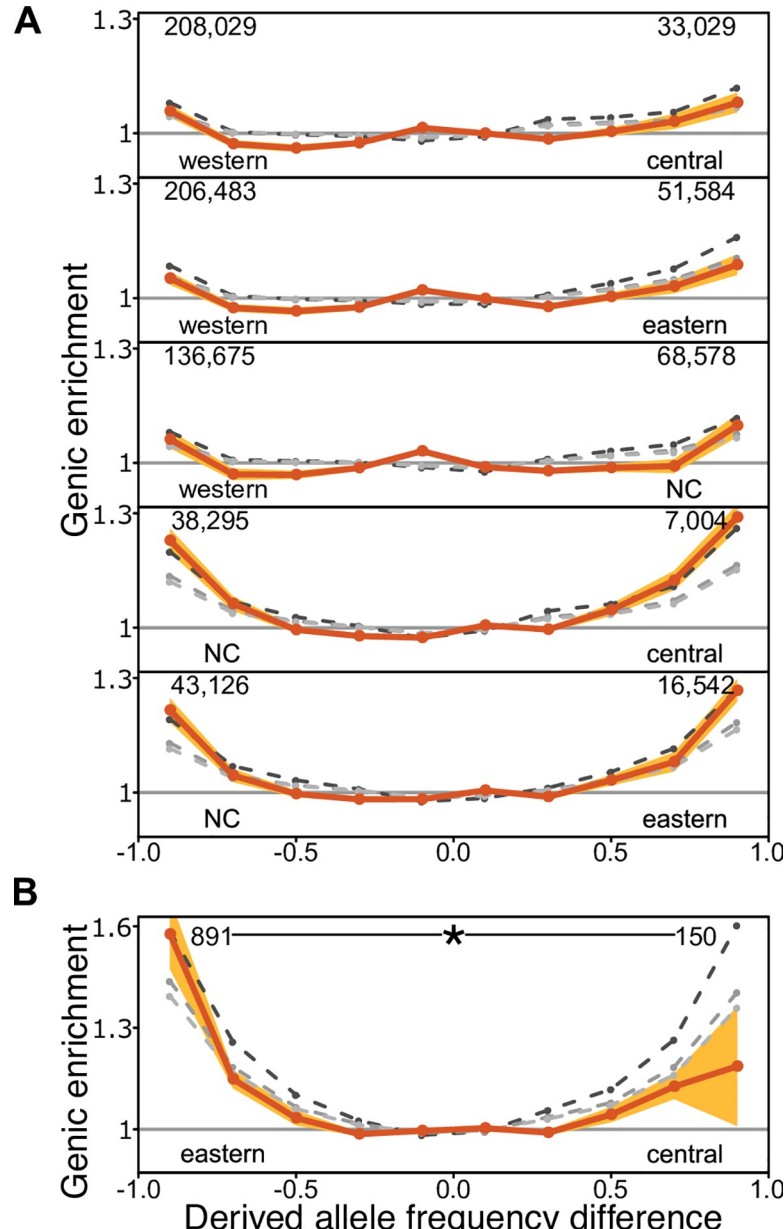

**Fig 2. Genic enrichment in bins of signed difference in derived allele frequency ($\delta$).** A, X-axis: $\delta$ is computed as the difference in derived allele frequency, for each pair of chimpanzee subspecies. Tail bins (the last bin in either end of $\delta$) contain those SNPs with the largest allele frequency differences. Numbers are of the genic SNPs in each tail bin. Y-axis: genic enrichment in each $\delta$ bin (Methods). B, Genic enrichment eastern and central chimpanzee $\delta$, plotted separately due to a different Y-axis limit. NC = Nigeria-Cameroon. The asterisk shows significance of the asymmetry in the genic enrichment (* = 0.01). Shading represents the 95% CI (i.e. alpha = 0.05 for a two-tailed test) estimated by 200kb weighted block jackknife. Grey dashed lines represent simulations under increasing levels of background selection that best match different aspects of the data: lightest to darkest shades: B = 0.93 (excluding $\delta$ tail bins), 0.92 (all $\delta$ bins), and 0.88 (unmodified genic B values form McVicker et. al. 2009 [34]).

enrichment is symmetric (Fig 2A) with symmetry defined as overlapping $\delta$ tail bin genic enrichment 95% CIs).

In marked contrast to these symmetric enrichments, we find a distinctive asymmetry between the tail bin genic enrichments of central and eastern chimpanzees (Fig 2B). The

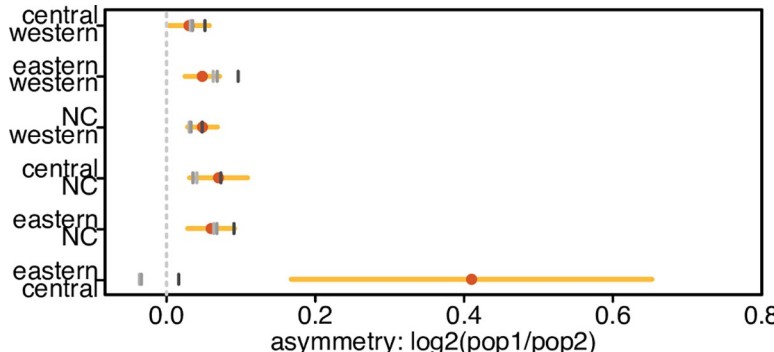

**Fig 3. Direct quantification of δ tail bin genic enrichment asymmetry.** The asymmetry of the genic enrichments in the δ tails is measured by taking their $\log_2$ ratio, thus 0 indicates a symmetric enrichment (equal enrichment in both δ tails). NC = Nigeria-Cameroon. Dot = observed asymmetry. Horizontal lines represent confidence intervals estimated by 200kb weighted block jackknife (light = 95%, i.e. alpha = 0.05 for a two-tailed test). Grey vertical marks represent the δ tail asymmetry in simulations, under increasing levels of background selection that best match different aspects of the data: lightest to darkest shades: B = 0.93 (excluding δ tail bins), 0.92 (all δ bins), and 0.88 (unmodified genic B values form McVicker et. al. 2009 [34]).

central δ tail exhibits a typical genic enrichment (1.19, 95 per cent confidence interval 1.01–1.36) but surprisingly, the eastern δ tail has a much stronger genic enrichment (1.58, 95 per cent confidence interval 1.47–1.68) that is significantly greater than the central tail (P < 0.005; weighted 200kb block jackknife, see Methods) and any other δ tail (all P < 0.0001; weighted 200kb block jackknife).

The large confidence interval of the central chimpanzee δ tail genic enrichment is due to the low number of SNPs in this tail. But we also identified a highly unusual 200kb genomic block on chromosome 3 (chr3:46508626–46708625) that contains 67 highly differentiated alleles between central and eastern chimpanzees, similarly distributed among the two tails (35 genic SNPs in the central tail and 32 genic SNPs in the eastern tail). Concerned that this block could bias our results, we repeated the enrichment analysis after excluding all SNPs contained within it. Removing this block reduces the genic enrichment slightly in the eastern tail (1.56) but substantially in the central tail (1.03) resulting in an even stronger asymmetry between the tails.

To directly quantify the asymmetry of the eastern and central chimpanzee δ tail genic enrichments, we tested if the $\log_2$ ratio of each pair of δ tail bin genic enrichments departs from zero, with the expectation that $\log_2$ ratios would be randomly distributed about zero if the δ tail genic enrichments are symmetric. Not surprisingly, we find a trend for the genic enrichment to be greater for the subspecies with the higher long-term $N_e$, although the $\log_2$ ratios are similar and small (ranging from 0.03 and 0.07). The only exception is for eastern vs. central, where it is 0.41 (95% CI, 0.17–0.65, 200kb weighted block jackknife). This is six times larger than the highest ratio between other subspecies pairs (Fig 3; S2 Table, Bonferroni corrected *p-value* δ western vs. central = 0.16 all other Bonferroni corrected *p-values* < = 0.004, *z-test*). The eastern vs. central asymmetry in genic enrichment is thus a clear outlier (*p-value* < 2.2e-16, two-sided Kolmogorov-Smirnov test).

## Recombination rate variation does not explain the δ tail asymmetry

Lower average recombination rates at genic versus non-genic sites may influence our analysis [30, 31], and we find a lower average inferred recombination rate at genic than non-genic sites (1.36 cM/Mb versus 1.61 cM/Mb) based on the existing, population-based chimpanzee

recombination map [32]. While this could lead to uniform increases in genic differentiation in genic sites, it is difficult to imagine that it would lead to qualitatively differences between sub-species, especially in the closely related eastern and central chimpanzees: while the recombination rate of individual sites may change through time, the global recombination rate at genic and non-genic sites will not differ among them. Indeed, low recombination rate does not drive our signal: recombination rates for both the eastern and central $\delta$ tails are not lower than the genic average recombination rate or that recombination rate in the other $\delta$ tails (S2 Fig), indicating that outliers in the recombination rate distribution do not contribute disproportionately to the $\delta$ tails in this pairwise comparison, in contrast to other pairwise $\delta$ tails, (S2 Fig). To further test the effect of varying recombination rates in the results, we repeated the $\delta$ tests restricting the analysis to sites with recombination rate < 0.5 cM/Mb. By restricting to this rate range, we equalise the average genic and non-genic recombination rates (0.16 cM/Mb versus 0.15 cM/Mb), and remove the association between $\delta$ and recombination rates. Still, all $\delta$ tails are significantly enriched for genic sites (S2 Fig). Most importantly, the eastern $\delta$ tail genic enrichment is significantly greater than that of central chimpanzees (S2 Fig), showing that differences in recombination rate do not drive the asymmetric genic enrichment observed in this comparison.

## Background selection does not explain the $\delta$ tail asymmetry

A certain level of $\delta$ tail bin genic enrichment (Fig 2) is, in principle, compatible with both recent positive selection and background selection (BGS) [10], the latter because linkage to sites under purifying selection reduces $N_e$ locally in genic regions and increases the effects of random genetic drift [33]. BGS in the range estimated for humans [34] can, for example, explain the $\delta$ tail bin genic enrichment in human populations, suggesting that this pattern is not evidence for pervasive recent human adaptation [10, 11, 29]. In this light, we next explored if BGS can explain our observed $\delta$ tail bin genic enrichments, and in particular the observed asymmetry between eastern and central chimpanzee $\delta$ tail bin genic enrichments.

The strength of BGS can be quantified as a B value, the ratio of diversity at a neutral site linked to sites under purifying selection compared to the expected neutral diversity in the absence of purifying selection. Equivalently, this can be expressed as the ratio of the respective $N_e$ [35, 36]. Previous attempts to simulate the effects of BGS on $\delta$ tail bin genic enrichments have simulated a single genome average B value [10, 11, 29], but the genome is heterogenous with regard to both the local density of sites under selection and the relative strength of selection, and thus B is expected to vary across the genome. We thus run coalescent simulations (Methods) with B values sampled from the empirical distribution estimated for the human genome by McVicker *et. al.* 2009 [34].

The B value that best explains the genic enrichments across all bins of $\delta$ (the B value that minimizes the summed square differences between observed and simulated enrichments across all pairwise $\delta$ bins, S2 Appendix) is 0.92 –i.e. a reduction of diversity of ~ eight per cent with respect to non-genic regions–decreasing to 0.93 (weaker BGS) when excluding the $\delta$ tail bins. We note that the vast majority of genic sites do not fall within the $\delta$ tail bins, thus a B of 0.93 explains most of the $\delta$ genic enrichment spectrum (Fig 2, lightest dashed line).

When fitting solely the twelve $\delta$ tail bins (Methods, S3 Table), 0.92 is still the best fitting B, followed closely by 0.90 with a 3.5% worse fit. This suggests that BGS that reduces diversity up to 10 per cent is enough to explain the observed $\delta$ tail bin genic enrichments. We note that simulations of BGS using a single genome average B value for all sites shows congruent results, but it requires much stronger BGS (B = 0.86) to match the 12 twelve $\delta$ tail bin genic enrichments (S2 Appendix, S2 Fig).

Previously, it was shown that BGS alone does not produce $\delta$ tail bin genic enrichment asymmetries in comparisons of human populations [11]. We also find that BGS does not result in significant eastern vs. central $\delta$ tail bin genic enrichment asymmetry. Simulations show a slight asymmetry in the tail genic enrichment (Figs 2B and 3) due to differences in their demographic histories (S2 Appendix, S4 Table). Nevertheless, no simulated value of B in the range (0.93–0.88) results in a tail genic enrichment log$_2$ ratio that falls within the 95% CI of the observed ratio (Fig 3). Further, we could not find a B value that results in a genic enrichment that lies within the 95% CI for both eastern and central chimpanzees. In contrast, the small (though statistically significant) asymmetries in all other pairwise $\delta$ tail bin genic enrichments are observed in simulations and thus fully explicable by demography and BGS (Fig 3). We note that the unmodified McVicker B values (which have an average B of 0.88) could explain the observed eastern $\delta$ tail bin genic enrichment, but they provide an extremely poor fit to all other $\delta$ bins (Fig 2)–including the central $\delta$ tail. We hypothesise that this is due to the B inference ignoring the effects of positive selection, which likely have a pronounced effect on the variance of B.

Previous work has shown that background selection varies little among the great apes [18]. Theory suggests that the diversity-reducing effect of BGS is independent of $N_e$, being determined by the distribution of fitness effects ($s$), except for the narrow range of $N_e * s = 1$ (Nam et al., 2017), while previous work suggests that more than 80% of deleterious mutations in chimpanzees have $N_e s \gg 1$ [37] Thus, the expectation is that the diversity reducing effect of BGS should be the same across all four chimpanzee subspecies. Indeed, we find comparable effects of background selection across subspecies: the relative reduction in neutral variation linked to genes is comparable amongst chimpanzee subspecies (S4A Fig), and neutral diversity has similar dependency on recombination rate and density of functional features across subspecies (with the exception of western chimpanzees, S4B Fig). Further, using a population genetic statistical model [38] we estimate the same reduction in neutral diversity due to background selection in each chimpanzee subspecies, at 11%, in the highest likelihood model (S3 Appendix, S5 Table). Thus, despite their differing demographic histories (Fig 1), the effects of BGS are very similar across each chimpanzee subspecies. This justifies using the same average strength of BGS across subspecies above. Nevertheless, to explore if our conclusions are robust to this assumption, we also modelled a greater strength of BGS in eastern chimpanzees (B = 0.825, the value which best matches the eastern $\delta$ tail bin genic enrichment) than in the other subspecies (B range 0.900–0.850), using a single genome wide average B value for each scenario. Stronger BGS in eastern chimpanzees does not produce an eastern central $\delta$ tail bin asymmetry as large as that observed in the genomes (log2 ratio range 0.120–0.146), further illustrating that BGS cannot explain the greater tail genic enrichment in eastern chimpanzees (S5 Fig). Rather, this is most likely a signal of recent adaptation.

## Population-specific branch lengths with PBSnj

Pairwise comparisons cannot determine which subspecies has changed. Direction, and therefore biological meaning, to allele frequency difference can only be garnered by assuming that derived alleles most often provide the basis for new adaptations. This approach is also limited by the collapsing of the shared history of lineages. For example, in the Nigeria-Cameroon vs. eastern comparison, 22% of the SNPs in the eastern $\delta$ tail are also in the central $\delta$ tail (for Nigeria-Cameroon vs. central comparison), whereas only 3.5% are highly differentiated to both Nigeria and central chimpanzees. Thus, $\delta$ summarises the allele frequency change across several parts of the phylogeny, limiting the biological interpretation of its tails.

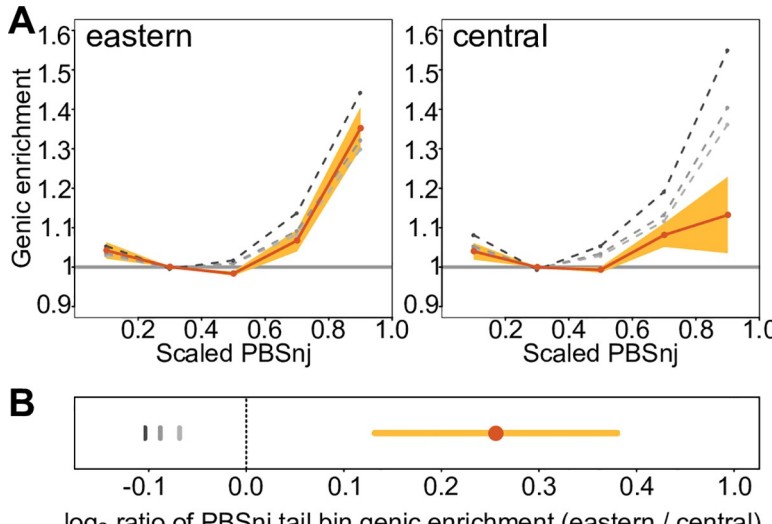

**Fig 4. Genic enrichment in bins of PBSnj in eastern and central chimpanzees**. **A** X-axes: PBS scaled to take values in the range 0–1. Y-axes: Genic enrichment computed as described in Fig 2. Shading represents the 95% CI (i.e. alpha = 0.05 for a two-tailed test) estimated by 200kb weighted block jackknife. **B:** $\log_2$ ratio of the eastern and central PBSnj tail (PBS > = 0.8) genic enrichment. **A,B** Grey dashed (**A**) or vertical (**B**) lines represent the PBSnj genic enrichment in simulations, under increasing levels of background selection that best match different aspects of $\delta$, as described in Figs 2 and 3: lightest to darkest shades: B = 0.93 (excluding $\delta$ tail bins), 0.92 (all $\delta$ bins), and 0.88 (unmodified genic B values form McVicker).

To overcome this limitation, we developed a statistic that extends the widely used Population Branch Statistic (PBS) [8]. Briefly, large PBS values identify targets of positive selection as SNPs with population-specific allele frequency differentiation, as these sites result in unusually long branch lengths in pairwise $F_{ST}$-distance trees between three taxa. Small PBS values are due to very short branches, for example due to purifying, shared balancing selection or rare mutations. We extend this test to more than three taxa in the novel *PBSnj* statistic by applying the Neighbor-Joining (NJ) algorithm on the matrix of the per-SNP pairwise $F_{ST}$ distances of the four subspecies (Methods, S4 Appendix). This way, PBSnj allows us to jointly compare the four subspecies and identify SNPs with very long branches (allele frequency differentiation) in one subspecies only. Additional advantages of PBSnj are that it does not rely on the specification of ancestral or derived states, and that the NJ algorithm does not require specification of a phylogenetic tree describing the relationship amongst taxa (S4 Appendix).

PBSnj allows us to determine within which lineage, eastern or central chimpanzees, allele frequencies have changed to result in the asymmetric $\delta$ genic enrichment. Analogous to the $\delta$ tail bins, we binned PBSnj scores and calculated the genic enrichment for each species PBSnj tail (Fig 4A). The PBSnj eastern tail has significantly stronger genic enrichment than the central tail (eastern: 1.35, central: 1.13, $\log_2$ ratio = 0.27, $p < 0.0005$ estimated from weighted 200kb block jackknife, Fig 4B). This shows that the central vs. eastern asymmetry in the $\delta$ tail bin genic enrichments (Figs 2B and 3) is due to the drastic allele frequency rise of genic SNPs in eastern chimpanzees since their divergence with central chimpanzees.

Again, recombination rate variation does not explain the observed eastern PBSnj enrichment, as it persists even after comparing genic and non-genic sites with similar recombination rates as described above (S6 Fig). Importantly, across the range of B values (0.88–0.93), simulations show that eastern and central chimpanzee PBSnj tail genic enrichments are expected to be equal or tend to be higher for central than eastern chimpanzees (Fig 4B). In fact, BGS would

need to be much stronger in eastern chimpanzees than in central chimpanzees to produce the observed levels of PBSnj tail genic enrichments. BGS with B < 0.93 would be required to produce the genic enrichment exhibited in the eastern PBSnj tail, but B > = 0.93 produces PBSnj tail genic enrichments of equal or greater magnitude as those seen for central chimpanzees, and it provides a much better fit to the data in the rest of the PBSnj distribution for both species (Fig 4A). This result is replicated in simulations with a single average B value for all sites, (S5 Appendix and S6 Appendix, and S6 Table). Thus, we observe strong evidence of positive selection for eastern chimpanzees: they exhibit the greatest genic enrichment for highly differentiated SNPs, an enrichment that (unlike in other subspecies) we cannot explain by demography and background selection alone. By using the genomic blocks used to estimate the PBSnj tail Confidence Intervals in Fig 4A, we estimate that an additional 12–23 population specific sweeps are sufficient to explain this signature (Methods, S7 Fig). Although this is a conservative estimate, it shows that we do not require an unrealistically large number of selective sweeps to explain the distinct pattern of eastern chimpanzees.

## PBSnj eastern tail SNPs have regulatory functions

While less than 1% of PBSnj eastern tail SNPs result in an amino acid change, compared to the genic background, PBSnj tail genic SNPs are significantly enriched in non-synonymous variants (PBSnj tail proportion amino acid change = 0.84%; genic background proportion = 0.06%, permutation test p = 0.001, S7 Table lists PBSnj eastern tail non-synonymous SNPs). This is not an indication that non-synonymous sites are especially important for local adaptation, but rather reflects an enrichment of PBSnj SNPs in exonic (PBSnj tail proportion = 1.84%; genic background proportion = 1.46%, permutation test p = 0.046) and CDS (PBSnj tail proportion = 3.97%; genic background proportion = 2.87%, permutation test p = 0.001) SNPs, and the observed proportion of exonic SNPs that are non-synonymous matches that of the genic background (PBSnj tail proportion 0.45, genic background proportion = 0.44, permutation test p = 0.53).

This puts the focus on regulatory changes. We used regulomeDB [39] to predict putative regulatory consequences of chimpanzee SNPs from the sequence context and biochemical signatures of homologous human sites. The PBSnj eastern tail genic SNPs are more likely to have strong evidence of regulatory function (3.7% vs. 3.0%, permutation test p = 0.012) and less likely to have no ascribed regulatory function (52.3% vs. 56.0%, permutation test p = 0.0001) than randomly sampled genic SNPs, S8 Table. In contrast, PBSnj central tail SNPs show no difference to the genic background for either category (S8 Table; Nigeria-Cameroon and western also exhibit weaker but significant enrichments). Interestingly, PBSnj eastern tail SNPs do not differ in functional constraint (as measured by *phastCons* scores [40], see Methods) from random genic SNPs (S9 Table). This suggests that while likely enriched in regulatory functions, these sites are not under particularly strong long-term constraint, perhaps because they do not affect functions that have been tightly conserved over long evolutionary times.

## Potential biological functions of the PBSnj eastern tail SNPs

To understand the biological mechanisms and putative selective factors driving the recent adaptations in eastern chimpanzees, we investigated the genes containing the genic SNPs in the PBSnj eastern tail (hereafter PBSnj eastern genes). As there are very few studies of the functional effects of chimpanzee genetic variation, we are reliant on homology and conservation of function to other primate species. We therefore tested three different gene sets that are potentially biologically informative: Human Gene Ontology (GO) annotations [41, 42]; human

Viral Interacting Proteins (VIPs) [4]; and genes with specific gene expression changes in the vervet monkey in response to SIV infection [43, 44].

Seven GO categories are significantly enriched (all $p <= 0.00026$, False Discovery Rate (FDR) < 0.1; GOWINDA; S10–S13 Tables), with a striking preponderance of immunity-related GO categories and genes involved in anti-viral activity. The top category is "cytoplasmic mRNA processing body assembly", and three of the five PBSnj eastern genes in this category (*DDX6* [45], *ATXN2* [46] and *DYNC1H1* [47]) are either key components of processing bodies (P-bodies) or regulate the assembly or growth of P-bodies in response to stress. Selection on the immune system is suggested also by the second category, "antigen processing and presentation of peptide antigen via MHC class I". The signal in this category is due to six genes, of which only *HLA-A* is an MHC gene, with the other genes being *B2M*, *ERAP1*, *PDI3*, *SEC13*, and *SEC24B*. There are three more significant categories related with immunity: "T cell co-stimulation", "negative regulation of complement-dependent cytotoxicity", and "type I interferon signalling pathway". Even the "cytoplasmic mRNA processing body assembly" category is potentially linked to virus infection as P-bodies are cytoplasmic RNA granules manipulated by viruses to promote viral survival and achieve infection [48, 49]. The enrichment in immune categories and virus-related genes is in perfect agreement with the PBSnj eastern genes being also enriched in three sets of VIPs [4]–genes with no annotated immune functions but that interact with viruses. Specifically, the enriched VIPs are for *Dengue virus*, *Bovine leukaemia virus* and *human T-lymphotropic virus* ($p < 0.01$, FDR < 0.1, see Table 1 and S14–S17 Tables), which are genes with no annotated immune functions but that interact with viruses.

The genes from these VIP sets are not members of the five significantly enriched immunity related GO categories, and VIP sets generally do not contain immunity genes. This provides an independent signal for the relevance of viruses to PBSnj eastern genes. Together, these results suggest that adaptation to pathogens, and viruses in particular, may have had an important role in the recent adaptation in eastern chimpanzees.

In light of the suggestive evidence for virus-related adaptation, our attention was drawn to the simian immunodeficiency virus. Amongst chimpanzee viruses, SIV is intensively studied as it is the progenitor of the human immunodeficiency virus (HIV) that created the global acquired immune deficiency syndrome (AIDS) pandemic. It is also of interest here because it appears to only infect natural populations of eastern and central chimpanzees [50–53], and because it has mediated fast, recent adaptations in other natural hosts [54]. Specifically, Svardal *et. al.* (2017) investigated a set of genes that change expression in response to SIV infection in SIV natural hosts (vervet monkeys) but not in non-natural hosts that develop immunodeficiency (macaques) [43, 44], hereafter referred to as "natural host SIV responsive genes". Natural host SIV responsive genes are likely involved in the specific immune response of natural hosts to SIV infection, which limits the effects of the virus and prevents subsequent immunodeficiency. These genes also show signatures of positive selection in vervet monkeys, suggesting that ongoing adaptation to the virus occurs even in natural hosts [54]. These observations also reveal natural host SIV responsive genes as a highly relevant candidate gene set for genes under positive selection in SIV infected species. Strikingly, the PBSnj eastern tail SNPs, but none of the other subspecies' PBSnj tails, are significantly enriched in these same natural host

**Table 1. VIP gene enrichment in the PBSnj eastern tail.**

| VIRUS | P-VALUE | FDR P-VALUE |
|---|---|---|
| BLV | 0.0015 | 0.0239 |
| DENV | 0.0025 | 0.0239 |
| HTLV | 0.0145 | 0.0780 |

**Table 2. SIV responsive gene enrichment in subspecies PBSnj tails.**

| Subspecies | Observed | Expected | P-VALUE |
|---|---|---|---|
| Eastern | 118 | 99 | 0.0198 |
| Central | 36 | 29 | 0.0739 |

SIV responsive genes [43, 44] (observed 118 genes, expected 100, *p-value* = 0.0195, GOWINDA, FDR = 0.064 see Methods, Table 2, S18 Table).

In fact, the set of natural host SIV responsive genes can fully explain the unique eastern signature: the asymmetry in the PBSnj tail is abolished when this set of genes is removed from the analysis–the genic enrichment in the eastern PBSnj tail decreases from 1.35 to 1.26, and the 95% confidence interval of this point estimate now overlaps those of Nigeria-Cameroon and central chimpanzees (Methods). A reduction in the genic enrichment in the PBSnj tail is expected, as it is enriched in natural host SIV responsive genes; but this exercise allows us to show that in the absence of selection in natural host SIV responsive genes, the signature of recent positive selection in eastern chimpanzees would not be exceptional.

The natural host response in vervet monkeys is associated with changes in the expression of these natural host SIV responsive genes. In agreement with potential adaptations in gene expression, the set of PBSnjE SNPs in the natural host SIV responsive genes are further enriched in sites with putative gene regulatory function (p = 0.0485 when compared with other PBSnj eastern tail genic SNPs, p = 0.0089 with all genic SNPs) and strongly depleted of sites with no predicted regulatory function (p = 0.0001 when compared with other PBSnj eastern tail genic SNPs, p = 0.0001 with all genic SNPs, S19 Table).

While these genes were not identified in chimpanzees where SIV experiments are naturally severely limited, this suggests a potentially similar mechanism of adaptation to SIV (or to an unknown virus with a similar effect in gene expression) in vervet monkeys and chimpanzees.

## Biological functions of the PBSnj central tail SNPs

Despite having a larger long-term $N_e$ than eastern chimpanzees, central chimpanzees do not show a clear genomic signature of recent adaptation. Despite being naturally infected by SIV and being the source of pandemic HIV, they show no clear indication of selection in SIV responsive genes: the PBSnj central tail has a greater number of SNPs in SIV responsive genes than expected (36 vs. 29), but the enrichment is non-significant (p = 0.0763; resampling test, Table 1). Power to identify a significant enrichment is hampered by the low number of SNPs. However, highly differentiated SNPs in the PBSnj long branches of central chimpanzees are significantly enriched in one GO category, "chemokine receptor activity", due to SNPs in *CCR3*, *CCR9* and *CXCR6* (p = 0.00001, FDR < 0.02, GOWINDA). Each of these genes is located within the large cluster of cytokine receptor genes on chromosome 3, but they appear to be associated with different sweep events (S8 Fig). Chemokine receptors facilitate the response to chemokine signalling, and their functions are important for both inflammatory and immunity responses [55]. For instance, *CCR3* is involved in the recruitment of eosinophils in the airway [56] and *CCR9* is implicated in T-cell development [57].

These genes are of also interest because the human paralogs *CCR5* and *CXCR4* are the two most common co-receptors for HIV-1 cell entry [58, 59]. Both *CCR3* and *CXCR6* can be used to enter the cell by some SIV, HIV-1 and HIV-2 subtypes [60–63], and the SIV of both sooty mangabey [64] and vervet monkey [65] use *CXCR6*. Although some results suggest that chimpanzee SIV cannot use chimpanzee *CXCR6* [66], the full breadth of co-receptors used by SIV in chimpanzees is still being investigated. For instance, it is unknown how important coding

sequence variation is to co-receptor function [66]. We note that one of the PBSnj tail SNPs in *CCR3* results in an amino acid substitution (246 S/A) in transmembrane domain 6, and the paralogous region has been implicated in the modulation of CCR5 activity [67]. Thus, changes in these co-receptors may have the potential to affect the entry of SIV in chimpanzee cells.

## Discussion

Comparing whole genomes from the four subspecies of chimpanzees we find that the alleles whose frequency rose quickly and substantially in particular chimpanzee subspecies, resulting in strong genetic differentiation, are enriched in genic sites. By simulating the effects of BGS, we show that most features of this genic enrichment can be explained by the demographic history of chimpanzees combined with BGS. However, different strengths of BGS are required to explain different aspects of the data. Weaker BGS, of B equal 0.93, best explains the genic enrichment for all but the most highly differentiated alleles among the subspecies. Stronger BGS, of B < = 0.92 is required to explain the genic enrichment in the most highly differentiated SNPs. This difference suggests a contribution of positive selection. Notably, the comparison of eastern to central chimpanzees shows an asymmetry in the genic enrichment that we cannot explain by BGS alone. Our PBSnj statistic shows that this signature is due to SNPs whose frequency have changed specifically in eastern chimpanzees since their divergence with central chimpanzees.

Many of these SNPs are polymorphic in central chimpanzees, so it is likely that many of these adaptations have occurred from standing genetic variation and consist of soft sweeps [68]. This would suggest that adaptation from standing genetic variation is important throughout primate evolution, not just in recent human evolution [69]. Alternatively, some of these sites may be polymorphic in central chimpanzees due to gene flow from eastern chimpanzees. The inferred chimpanzee demography includes recurrent migration between eastern and central chimpanzees, in both directions ([20] and see Methods), indicating that selection in eastern chimpanzees was strong enough to overcome the homogenising effect of gene flow from central chimpanzees.

These strongly differentiated alleles in eastern chimpanzees are enriched in sites with inferred regulatory function, but not in sites that have been strongly constrained during mammalian evolution. This agrees well with a role in adaptation to pathogens, which is often characterized by fast arms-race evolution. The PBSnj eastern genes are enriched in several immune-related categories, with many of them having known or potential virus-related functions. *OAS2* and *RNASEL*, for example, are involved in foreign RNA degradation [70], while *ERAP1* is a gene under long term balancing selection in humans [71] that is involved in MHC class I epitope presentation [72]. These are plausible adaptations to viral infections in eastern chimpanzees. In fact, these PBSnj eastern sites are located disproportionately in genes that differentiate the CD4 transcriptional response to SIV in a natural host species that tolerates the virus from a non-natural host species that develops immunodeficiency. Selection acting on this set of genes is sufficient to produce the greater eastern signal. Two aspects of this enrichment are notable. First, these genes are identified based on gene expression responses in vervet monkeys and macaques to SIV infection [43, 44], and are thus completely independent of chimpanzee genetics. Second, the SIV responsive genes also show diversifying selection in vervet monkeys [54]. Of note, the PBSnj eastern SNPs are strongly enriched in putative regulatory functions, in agreement with putative adaptations through gene expression.

How may this happen? The genes that are both SIV-responsive and contain PBSnj eastern tail SNPs are significantly enriched in four GO categories (FDR < 0.1, GOWINDA, S21 Table). The top category is "type I interferon signalling pathway with four genes (*IRF2*,

*RNASEL*, *HLA-A* and *SP100*). This category is also significantly enriched in the full set of PBSnj eastern tail SNPs. *OAS2* is also in this category but it is inducible in both vervet and macaque shortly after SIV infection. *IRF2*, *RNASEL* and *SP100* are all upregulated in the CD4 cells of vervet monkey but not of macaques one day post infection. This is relevant as regulation of the interferon response is a key differentiator between natural and non-natural SIV hosts [73] and the timing of interferon responses can be key in the progression to AIDS in humans infected with HIV [74, 75]. Another enriched category is "polycomb group (PcG) protein complex". PcG complexes can be involved in the epigenetic regulation of HIV-1 latency [76, 77], and three of the genes in this GO category, *PHC2*, *CBX7* and *KDM2B* encode components of the same PcG complex, PCR1 [77].

Of course, it is also possible that other viruses elicited a selection response in eastern chimpanzees, and in particular the SIV signature that we observe could be due to selection by other ssRNA viruses. Possibilities include the viruses involved in the three significant sets of VIPs, which are *Dengue virus* and the closely related *Bovine leukaemia virus* and *human T-lymphotropic virus*. However, we argue that SIV is a better candidate to explain our observations. Aside from the evidence for ongoing selection of these genes in vervet monkeys [54], there are some suggestions that SIV infection decreases fitness in populations of eastern chimpanzees infected with the virus [26], with some infected individuals described as having an AIDS-like pathology. It is thus plausible that the virus is a selective force in this subspecies. It is interesting that while the eastern PBSnj tail is enriched in SIV-response genes, it is not enriched in the SIV, HIV-1 or HIV-2 VIP categories (S14 Table). Further, while some VIPs are also SIV-response genes, neither SIV, HIV-1 nor HIV-2 VIPs are more likely to be SIV responsive genes than expected by chance (two-tailed binomial tests, number of genes in VIP category: 32 HIV-1, 8 HIV-2 SIV, observed proportions: HIV-1 0.13, HIV-2 0.38, SIV 0, expected proportions = 0.13, *p-values*: HIV-1 = 0.77, HIV-2 = 0.07, SIV = 1). Thus, these results suggest that the putative eastern chimpanzee-specific adaptations to SIV are mediated by expression changes in factors that modulate the immune response to infection, rather than changes in expression levels of direct SIV/HIV interactors. Our inferences are based on statistical enrichments and gene function, and additional work would be necessary to definitively show the effect of species-specific variants in function and phenotype. This is extremely challenging as chimpanzees are, like all other great apes, endangered and strongly protected. Nevertheless, the combination of results pointing to SIV-related genes suggests that SIV exerts a selective force in chimpanzees; together with work in other species [54] it also suggests that SIV is likely an important selective force in several natural primate species, which both vervet monkeys and eastern chimpanzees may respond to by shaping gene expression. Thus, while we cannot be completely certain that SIV is driving selection in eastern chimpanzees, this virus is the best candidate considering all currently available evidence.

It is also probable that eastern chimpanzees have adapted to additional selective pressures unrelated of viral pathogens or immunity. An obvious candidate would be life history traits. For example, the gene *SKOR2*, which contains the fifth ranked eastern specific missense polymorphism, has been associated with the timing of female puberty in GWAS of age of menarche in humans [78]. Unfortunately, like most polygenic traits, the genetic basis of menarche is poorly understood making it hard to contextualise this result.

Perhaps surprisingly, central chimpanzees have weaker signatures of natural selection despite being the subspecies with the largest $N_e$ [1, 20]. A few factors could blunt the evidence for positive selection in central chimpanzees, but none of them are able to explain the observed difference in PBSnj tail genic enrichment between central and eastern chimpanzees–including putative population substructure, gene flow from eastern chimpanzees or introgression from bonobos (S7 Appendix). Central chimpanzees also do not have significant enrichment in SIV

responsive genes despite, like eastern chimpanzees, being naturally infected by SIV [28]. However, central chimpanzees exhibit a significant enrichment of highly differentiated SNPs in the chemokine genes *CCR3*, *CCR9* and *CXCR6*. Given the known functions of these chemokine receptors [55] this result is suggestive of immunity related selection in central chimpanzees. In particular, we note that *CCR3* and *CXCR6* are used by SIV, HIV-1 and HIV-2 subtypes [60–65]. The signature of positive selection in *CXCR6* is interesting because the SIV of natural hosts sooty mangabey [64] and vervet monkeys [65] predominantly use *CXCR6* for host cell entry. This is in contrast with the dominant *CCR5* usage in hosts such as humans and macaques that progress to AIDS. While it is unclear which particular channels are used by SIV in each chimpanzee subspecies [66], the evidence of selection in central chimpanzees in these receptors raises the intriguing possibility that the two chimpanzee hosts have in part used distinct evolutionary responses to the virus: limiting cell entry in central chimpanzees; modulation of gene expression response in eastern chimpanzees.

While our attention has focussed on eastern, and to a lesser extent, central chimpanzees, this is not to say that positive selection has not acted on western and Nigeria-Cameroon chimpanzees. By applying a statistical model [29] to neutral diversity as a function of distance to genes we find that while chimpanzees share the same average strength of BGS, the reduction in neutral diversity due to hard selective sweeps is inferred to be greatest in western chimpanzees and least in central and eastern chimpanzees (S6 Table). This is in contrast to our main results for $\delta$ tail bin and PBSnj genic enrichments. This contradiction can be resolved in part if we posit that the rate of soft selective sweeps also increases with $N_e$ [18], as soft sweeps will not have the same diversity reducing effect as hard sweeps. Nevertheless, regardless of the existence of any relative differences in the type of selective sweeps, the divergence times and high genetic drift of the western and Nigeria-Cameroon lineages makes tests of allele frequency differentiation less well suited to identify adaptive loci than in eastern and central chimpanzees. Alternative approaches, for example using intensive within subspecies sampling, can help identify adaptive loci in these subspecies. Nonetheless, our results show striking differences between the sister subspecies of eastern and central chimpanzees. Besides helping us start to identify the genetic and phenotypic differences among subspecies, this finding highlights the need for genetic studies and conservation efforts to account for functional differentiation between subspecies and local populations across the entire chimpanzee range.

## Materials and methods

### Genotypes, haplotypes and genic regions

We analysed the 58 chimpanzee genomes described in de Manuel *et*. *al*. (2016), with sample sizes of: eastern 19, central 18, Nigeria-Cameroon 10, western 11 after excluding the hybrid Donald. For most tests based on allele frequencies, we used the chimpanzee VCF file from de Manuel et al., (2016) after removing every SNP with at least one missing genotype across all chimpanzees. For haplotype phasing, we also included the 10 bonobo genomes from [20]. To statistically phase haplotypes we used *Beagle [79] v 4.1* (downloaded from https://faculty.washington.edu/browning/beagle/b4_1.html, May 2016). We used default parameters without imputation, except that after the initial 10 burn in iterations we performed 15 phasing iterations (default is five) using the following command line: java -Xmx12000m -jar beagle.03-May16.862.jar gt = vcf out = vcf.phased impute = false nthreads = 1 niterations = 15

For the analysis of $\delta$ we chose to use the inferred ancestral chimpanzee allele from EPO alignments from ensembl. For comparison, and to show that our result was robust to AA inference method we also used the homologous human genome reference allele as the ancestral state for chimpanzee SNPs. We used the human genome from the 1000 Genomes project

phase III human_g1k_v37.fasta, available from: http://ftp.1000genomes.ebi.ac.uk/vol1/ftp/technical/reference/human_g1k_v37.fasta.gz

We used the UCSC liftover utility to convert chimpanzee SNPs' coordinates from pantro 2.1.4 to human genome version 37 (hg19) coordinates, then used samtools faidx to retrieve the human allele for that position. Both of these inference methods recovered the same signal of a significantly greater $\delta$ tail bin genic enrichment for eastern vs. central chimpanzees, see S22 Table. Again, we also note that our new statistic PBSnj does not require inference of the ancestral allele. In total we analysed 29,778,147 autosomal SNPs. Of these there was EPO high confidence inferred ancestral states for 27,081,963 SNPs and human inferred ancestral states for 28,776,489 SNPs.

We considered protein-coding genes on the autosomes (17,530 genes) and define 'genic sites' by extending the transcription start and end coordinates from ENSEMBL biobank for pantro2.1.4 by 2kb on each side.

### Genetic map

For statistics that required a genetic map, we used the pan diversity genetic map [32] inferred from 10 western chimpanzees. We downloaded the chimp_Dec8_Haplotypes_Mar1_chr-cleaned.txt files from birch.well.ox.ac.uk/panMap/haplotypes/genetic_map. These files consist of SNPs and their inferred local recombination rate. These map data were inferred from sequences aligned to the pantro2.1.2 genome, so we used two successive liftover steps to convert the coordinates of sites used to infer the genetic map to pantro2.1.4 coordinates: pantro2.1.2 to pantro2.1.3, then pantro2.1.3 to pantro2.1.4. Two steps are required as there are no liftover chains relating pantro2.1.2 to pantro2.1.4. Of the 5,323,278 autosomal markers, 33,263 were not lifted from pantro2.1.2 to pantro2.1.3. The remaining 5,290,015 were also successfully converted to pantro2.1.4 coordinates. After liftover we filtered sites that after the two steps were mapped to unassigned scaffolds or the X chromosome, which left 5,289,844 SNPs. Next, we sorted loci by position to correct cases where their relative order was scrambled. This left a final number of 5,289,460 autosomal SNPs. Recombination rates were then recalculated by linear interpolation between consecutive markers (marker x, marker y) using the average of their estimated recombination rates (rate x, rate y).

### Signed difference in derived allele frequency ($\delta$)

Using the derived allele frequency of each SNP for each subspecies we calculated, for each pair of chimpanzee subspecies, the signed difference in derived allele frequency (DAF) between them: $\delta = DAF_{pop1} - DAF_{pop2}$; $DAF_{pop1} > DAF_{pop2}$: $\delta > 0$; $DAF_{pop1} < DAF_{pop2}$: $\delta < 0$; $-1 <= \delta <= 1$. We bin $\delta$ into 10 bins of 0.2. The choice of subspecies assigned to pop1 or pop2 is arbitrary and has no bearing on the results. To ensure that both tail bins are identically wide, we define them as Bin 1: $-1 <= \delta <= 0.8$ and Bin 10 as $0.79 < \delta <= 1$. As a consequence, the Bin 5 ($0.00 < \delta < 0.2$) is marginally narrower than the other bins (by 0.01), but it contains a large number of sites and the slight size difference has negligible impact on the analyses.

We estimate confidence intervals and infer *p-values* for $\delta$ genic enrichment using a weighted block jackknife [80] utilising the method of Busing *et. al.* [81]. This has been used for analogous tests, as it accounts for linkage disequilibrium, which means that SNPs in $\delta$ bins are not full independent of each other. We divide the genome into non-overlapping 200kb windows to capture the blocking effect of LD. We then recalculate, for each bin, the genic enrichment using a delete-1 window jackknife. We also weight the windows by the total number of SNPs in them, to downweigh, within each bin, blocks with large numbers of linked SNPs. We determine that two tails are differentially enriched if their 95% CIs of enrichment do not

overlap. For directly testing asymmetry (or in the case of PBSnj, equality) using the $\log_2$ ratio, we use the same weighted block jackknife, and use the 95% CI as a two-tailed test with alpha = 0.05. Other enrichment and resampling tests are described in Methods subsection "Statistics".

## Population Branch Statistic neighbour-joining

The Population Branch Statistic (PBS), [8] is a test of population specific natural selection. In the framework of a three-taxon distance tree, SNPs under selection specific to one population are detected as those that result in longer than expected branch lengths (large allele frequency differentiation). To generate the tree, for each site, the full distance matrix of pairwise $F_{ST}$ is computed. A three taxa tree is unrooted and has only one possible topology, so simple algebra allows the calculation of each branch length in the tree. Extreme outliers in the distribution of PBS are considered candidates of positive selection.

We introduce Population Branch Statistic neighbour-joining (PBSnj) as a simple method to calculate population specific branch lengths when more than three taxa are being analysed. We note that related Methods have recently appeared in the literature [82, 83]. Full details are in S4 Appendix, but in brief, using the full matrix of pairwise $F_{ST}$, $F_{ST}$ values are transformed to units of drift time as $\ln (1-F_{ST})$ [8]. For fixed differences this transformation is mathematically undefined i.e. $\ln (0)$, and $F_{ST} = 1$ is replaced with the next largest observed $F_{ST}$ value for a given population pair. Then the Neighbor-Joining algorithm [84] is used to infer the tree topology and calculate branch lengths. This overcomes errors in the inferred length of external branches due to misspecification of a fixed tree topology. To enable a binning scheme of PBSnj values that is comparable between subspecies, these scores are further normalised to be on the 0–1 scale.

$F_{ST}$ for PBSnj was calculated using the estimator described in [85] because there are unequal sample sizes for the subspecies, and the classical Weir and Cockerham estimator can be biased with unequal sample sizes [85]. To calculate genic enrichments along the PBSnj distribution we bin SNPs in PBSnj bins 0.2 units wide. As for $\delta$ analyses, we use the 200 kb weighted block jack-knife to estimate confidence and significance levels. We provide a source code file, written in R, to calculate PBSnj ("PBSnj_function.R"), see Data availability.

## Model of Chimpanzee demographic history

The most detailed exploration of chimpanzee demography comes from the work of de Manuel *et. al.* (2016). This paper describes the 58 chimpanzee full genome sequences we use here, and estimation of their inferred demographic model. As this paper took a primary interest in investigating chimpanzee-bonobo post speciation gene flow, and to reduce the number of parameters to be estimated, models were inferred using either Nigeria-Cameroon or western chimpanzees, but not both. Thus, de Manuel *et. al.* (2016) provides "bonobo, eastern, central, Nigeria-Cameroon" and "bonobo, eastern, central, western" models. These are referred to, respectively, as 'becn' and 'becw' models below.

For this investigation we use a merged demographic history. To begin the construction of this model, we recognised that there is little gene flow involving western chimpanzees in the 'becw' model, but that gene flow events are a key determinant of patterns of chimpanzee genetic diversity and differentiation in the 'becn' model. We therefore used the 'becn' model as a scaffold to which parameters relating to western chimpanzees (bottlenecks, expansions and $N_e$ estimates) from the 'becw' model are "grafted" in, to create a merged 'becnw' model. To make sure that the $N_e$ of western chimpanzees was appropriately scaled, all $N_e$s 1000 ya pastwards for western chimpanzees specified in the 'becw' model were normalized by multiplying

by the ratio of the inferred $N_e$s of central chimpanzees specified from 1000 ya pastwards in the 'becn' and 'becw' models: scaled western $N_e$ = western $N_e$ * 3.66914400056 / 4.3158739382. Present western $N_e$ was normalised by the ratio of the present central $N_e$: scaled western $N_e$ = western $N_e$ * 0.3092 / 0.30865.

Initially, we used the split time of the western and Nigeria-Cameroon lineages of 250ky reported by de Manuel *et. al.* which was estimated from sequence divergence data, but this gave a bad fit to $F_{ST}$ values, being substantially lower than observed (S23 Table). We addressed this by increasing the western/Nigeria-Cameroon divergence time in proportion to the ratio of model:observed western/Nigeria-Cameroon $F_{ST}$. i.e. $F_{ST}$Observed / $F_{ST}$Model = timeX / 250kya => timeX = $F_{ST}$Observed / $F_{ST}$Model x 250kya. We adjust the observed $F_{ST}$ by -0.008 –to capture the average difference between model versus observed $F_{ST}$ values for central/eastern/Nigeria-Cameroon chimpanzees. This simple calculation results in an adjusted time of 267kya for the western/Nigeria-Cameroon split. $F_{ST}$ values for this new model show a much better fit to observed values (S23 Table), and it is this model that we use for all subsequent modelling of genic enrichments and the effects of background selection.

To determine model fit above, we calculated all pairwise average $F_{ST}$ values for the simulated data and compared them to the empirical $F_{ST}$ estimates. For each scenario, we simulated 1,000,000 2kb fragments (2 Gb of sequence).

All simulations of neutral diversity and background selection were performed with *msms* [86], and following de Manuel *et. al.* assuming a mutation rate of $1.2e^{-8}$ and recombination rate $0.96e^{-8}$, with the following command line:

msms 116 1 -t 0.96048 -r 0.768384 2001 -I 5 0 38 36 20 22 0 -n 1 0.0742 -n 2 0.3181 -n 3 0.3092 -n 4 0.0386 -n 5 0.08114434 -m 1 2 0 -m 1 3 0 -m 1 4 0 -m 2 1 0 -m 2 3 1.8181960943074 -m 2 4 0 -m 3 1 0 -m 3 2 2.02290154800773 -m 3 4 0 -m 4 1 0 -m 4 2 0 -m 4 3 0 -m 5 1 0 -m 1 5 0 -m 5 2 0 -m 2 5 0 -m 5 3 0 -m 3 5 0 -m 4 5 0 -m 5 4 0 -en 0.001 1 1.83290809268 -en 0.001 2 1.161030985567 -en 0.001 3 3.66914400056 -en 0.001 4 1.23640124358 -en 0.001 5 0.9132505 -em 0.020875 1 2 0 -em 0.020875 1 3 0 -em 0.020875 1 4 0 -em 0.020875 2 1 0 -em 0.020875 2 3 1.8181960943074 -em 0.020875 2 4 1.12888460726286 -em 0.020875 3 1 0 -em 0.020875 3 2 2.02290154800773 -em 0.020875 3 4 0.514005225416364 -em 0.020875 4 1 0 -em 0.020875 4 2 0.61034918826118 -em 0.020875 4 3 2.77081002950074 -em 0.042025 1 2 0 -em 0.042025 1 3 0.0447270935214584 -em 0.042025 1 4 0.00204350937063846 -em 0.042025 2 1 0 -em 0.042025 2 3 1.8181960943074 -em 0.042025 2 4 1.12888460726286 -em 0.042025 3 1 0.0340892941439601 -em 0.042025 3 2 2.02290154800773 -em 0.042025 3 4 0.514005225416364 -em 0.042025 4 1 0.00878072013784504 -em 0.042025 4 2 0.61034918826118 -em 0.042025 4 3 2.77081002950074 -en 0.104325 2 0.0402577179646081 -en 0.104325 3 0.192594746352967 -en 0.106325 3 8.73162876459514 -ej 0.106325 2 3 -em 0.106325 1 2 0 -em 0.106325 1 3 0.0177338314347154 -em 0.106325 1 4 0.00204350937063846 -em 0.106325 2 1 0 -em 0.106325 2 3 0 -em 0.106325 2 4 0 -em 0.106325 3 1 0.00723425109237692 -em 0.106325 3 2 0 -em 0.106325 3 4 0.193855714034029 -em 0.106325 4 1 0.00878072013784504 -em 0.106325 4 2 0 -em 0.106325 4 3 0.00771007640703268 -en 0.21195 5 0.1223036 -en 0.214175 5 0.194964 -en 0.267475 4 1.23640124358 -en 0.267475 5 0.194964 -ej 0.2675 5 4 -en 0.41955 1 0.158405393915496 -en 0.42155 1 0.299481445247702 -en 0.473075 4 0.0306317427630759 -en 0.475075 4 2.79429564470655 -en 0.480625 4 0.0872103733618782 -em 0.480625 1 2 0 -em 0.480625 1 3 0.0177338314347154 -em 0.480625 1 4 0.00204350937063846 -em 0.480625 2 1 0 -em 0.480625 2 3 0 -em 0.480625 2 4 0 -em 0.480625 3 1 0.00723425109237692 -em 0.480625 3 2 0 -em 0.480625 3 4 0.193855714034029 -em 0.480625 4 1 0.00878072013784504 -em 0.480625 4 2 0 -em 0.480625 4 3 0.00771007640703268 -en 0.482625 3 1.66920782430592 -ej 0.482625 4 3 -em 0.482625 1 2 0 -em 0.482625 1 3 0.241282075772286 -em 0.482625 1 4 0 -em 0.482625 2 1 0 -em 0.482625 2 3 0 -em 0.482625 2 4 0 -em 0.482625 3 1 0.0101771164248256 -em 0.482625 3

2 0 -em 0.482625 3 4 0 -em 0.482625 4 1 0 -em 0.482625 4 2 0 -em 0.482625 4 3 0 -en 1.5988 3
0.00336130452736601 -en 1.6008 3 1.47105091660349 -ej 1.6008 1 3 -em 1.6008 1 2 0 -em
1.6008 1 3 0 -em 1.6008 1 4 0 -em 1.6008 2 1 0 -em 1.6008 2 3 0 -em 1.6008 2 4 0 -em 1.6008 3
1 0 -em 1.6008 3 2 0 -em 1.6008 3 4 0 -em 1.6008 4 1 0 -em 1.6008 4 2 0 -em 1.6008 4 3 0

As a further assessment of the fit of the model, we plotted the observed and simulated site frequency spectrum (SFS), S9 Fig. In general, the model fit is good, being poorest for singletons (too high) and high frequency derived sites (too low). This is likely due to effects of selection on the genome, which is not incorporated into the neutral demographic model. We note too, that this model was computed using only the allele counts from regions of the genome under weak/no selection as inferred from GERP scores, further explaining the reduced fit at these two site classes.

Simulations of chimpanzee genetic data under neutrality and background selection.

We used *msms* to perform coalescent simulations of chimpanzee demography. To simulate the effects of background selection (BGS) we modified the estimates of effective population size ($N_e$) from the demographic model by multiplying them by a scaling factor, which represents the B score or effective reduction in $N_e$ due to BGS. 0.8, for example, reduces the $N_e$ and hence expected neutral diversity to 80% the level seen for neutral sites unlinked to regions under purifying selection [11]. To capture the possible variance in B across the genome, we sampled from the estimates of B for the human genome inferred by McVicker *et. al.* [34] downloaded from: http://www.phrap.org/software_dir/mcvicker_dir/bkgd.tar.gz. The coordinates for this file are for human reference genome assembly version hg18, and we used the coordinates of all annotated autosomal protein coding genes for hg18, +- 2kb, downloaded from ensembl.

McVicker *et. al.* estimated that autosomal diversity levels were reduced by ~ 20 per cent [34]. The mean B for genic regions is 0.75 and for non-genic regions B is 0.85, resulting in an average ratio (or effective genic B) of ~ 0.88, implying that diversity in genic regions should be reduced 12 per cent compared to non-genic regions. To shift this average B and explore the fit to the genomic data, we added a fixed constant to all genic B values, so that we could simulate BGS with different genome average B in the range 0.88–0.93, in 0.01 increments, in order to ascertain which average B best fit the observed $\delta$ genic enrichments. B scores were constrained to a maximum value of one, see S8 Fig for the distribution of non-genic and genic B scores. For non-genic regions and for each average B, we simulated 10 million 2.0 kb loci. After processing and calculating allele frequencies, we performed $\delta$ and PBSnj genic enrichments as described previously. To estimate a BGS strength that best matched the observed $\delta$ genic enrichments, we performed a simple sum of squared differences, summed for each $\delta$ genic enrichment bin for each pairwise comparison.

We also report results from simulations utilising a single genome average B value for all genic sites. We simulated non-genic regions with B = 1, and genic regions with different chosen strengths of BGS. We used B in the range 1–0.8, incremented by 0.025, with additional 0.0125 increments between 0.9–0.85. For neutral regions and for each B we simulated 25 million 2.0 kb loci. These serve a useful comparison to the simulations generated using McVicker derived B values.

## Estimating the number of extra eastern chimpanzee adaptive events

We use the structure of the block jack-knife to estimate the number of adaptive events that are needed to result in the PBSnj eastern tail genic enrichment being greater than that of central chimpanzees or generated by BGS. Recall that to estimate the error variance on the genic enrichment in each bin of PBSnj, we divided the genome into non-overlapping 200 kb blocks.

For each block we have the number of genic and non-genic SNPs per bin of the PBSnj distribution. For eastern chimps, there are 3528 genic SNPs contained within 832 blocks (i.e. 166 MB) in the PBSnj rhs tail i.e. with a PBSnj scaled length > = 0.8. NB: the majority of SNPs in this bin are not fixed, so we are not assuming that these are completed sweeps.

Of these, there are 448 blocks containing only 1 SNP i.e. 54% of blocks, 81 blocks with 10 or more outlier genic SNPs. i.e. 10% of blocks, with a block maximum count of 117 genic SNPs (S7A Fig).

We rank blocks by the number of genic SNPs that are outliers. Iterating over this sorted list we remove blocks and recalculate the enrichment for genic SNPs. We define matching as the number of iterations required to reduce the tail bin genic enrichment to below a magnitude less than the upper 95% confidence limit or the point estimate of the central PBSnj tail genic enrichment. We chose to order by the number of eastern tail genic SNPs as this results in a monotonically decreasing genic enrichment with each block being removed.

### Measures of conservation and effects on gene regulation

We used *phastCons* [40] to infer highly conserved sites. We used the 20 mammalian multiz alignment to the human genome hg38, downloaded from UCSC (http://hgdownload.cse.ucsc. edu/goldenPath/hg38/multiz20way/maf/). To reduce the chance that polymorphism in chimpanzees affects inference of conservation, we removed both the chimp and bonobo reference genomes form these alignments. We estimated the phylogenetic models from fourfold degenerate (non-conserved model) and codon first position sites (conserved model). We then predicted base conservation scores and conserved fragments using the following options:—target-coverage 0.25—expected-length 30. Resultant conserved elements covered 69.24% of the human exome, or an enrichment of 17.27. We note that although we attempted to remove the *Pan* branch from our estimates, it is impossible to completely avoid the use of these genomes, for example, when converting predicted conserved elements from hg38 to pantro2.1.4. These results have been deposited on Dryad (see Data availability).

We used regulomeDB [39] to identify putatively regulatory role of genomic sites. Due to the close phylogenetical relationship between chimpanzees and humans, we argue that in lieu of any functional data for chimpanzees, inferred function from homologous positions in the human genome is a useful proxy for function in the chimpanzee genome. To obtain regulomeDB information for variable chimpanzee positions we used liftover to map SNP coordinates from pantro2.1.4 to hg19, keeping positions that reciprocally mapped to homologous chromosomes. Alan Boyle then kindly provided regulomeDB annotations for these positions. In regulomeDB, lower scores reflect higher confidence in regulatory function. We modified scores on the basis that scores 1a-f are given for positions that are human eQTLs, which we do not use as they refer to the specific allele change in humans rather than to the function of the site. Without eQTLS, scores 1a-c and 2a-c reflect the same biochemical signatures and location within transcription factor motifs. Thus, we combine these scores in to a new "high confident" regulatory function category. Our "non-regulatory" category includes positions with regulomeDB scores of 6 or 7, which have no evidence of being regulatory. We did not use sites with intermediate scores.

### Gene set enrichment analyses

We used GOWINDA [87] to test for enrichments in Gene Ontology (GO) categories, which corrects for clustering and gene length biases. We used either GO categories or custom gene lists as candidate gene sets. GO categories for humans were obtained from the GO consortium [41, 42], while gene sets were manually created from published sets of Viral Interaction

Proteins [4] and a set of genes that are differentially expressed in CD4 cells after SIV infection in the natural SIV host vervet monkey but not in that non-natural host macaque [43, 44, 54].

GOWINDA has an input file format which enables flexible usage of nonstandard gene sets. Genes are defined in a gtf file. We created a gtf from the ENSMBLE gene definitions, but restricted these to genes with clear 1–1 orthologs with humans. Our gtf file contained 16,198 of 17,530 protein coding genes. This gene set has been deposited on Dryad (see Data availability). Additional inputs are the PBSnj tail SNP set, and the background SNP set (of which the candidates are a subset). For all gene set enrichments, the background SNPs set was the full genome-wide set of genic variants for which PBSnj could be calculated.

GOWINDA was designed to reduce false positives that result from gene length bias (the probability of randomly containing an outlier SNP increases with gene length) and clustering of genes (such as paralogs) that share function. It achieves this by using resampling of background SNPs, which is the genome wide set of SNPs considered in a test. We use the—mode *gene* switch. In this case, background SNPs are randomly sampled until the number of overlapping genes matches the total number of genes overlapping the PBSnj tail SNP set. Empirical *p-values* are estimated for each GO category, as the proportion of resamples which contain the same or greater number of genes than the PBSnj tail SNP set, per GO category (for each random background sample a pseudo *p-value* per GO category is also likewise calculated). FDR at each *p-value*, *p*, is then estimated as the number of observed *p-values* less than or equal to *p*, $R_{obs}$, divided by the total number of resamples with a *p-value* less than *p* $R_{exp}$ i.e. FDR = $R_{obs}$ / $R_{exp}$.

It is important to note that only genic background SNPs that are within the candidate set of genes (e.g. genes with GO definitions) are used in the random sampling. For the GO enrichment, after filtering for gene sets with at least 3 genes, the GO definition file contains definitions for 15649 genes, and 95% of genic background SNPs are used for resampling. This is important, as therefore GOWINDA cannot be used to directly test for enrichment in a single or small set of candidate gene sets. Providing one category, for example, would reduce the background SNP set to only those background SNPs in the genes in that category. Resampling can only ever return the same number of genes in this case. Thus, for VIPs and for the SIV gene set, we included an additional category, which is the full set of genes in the gtf file ("all gene set"). This has no effect on empirical *p-value* estimation. Its effect on FDR correction is limited as $R_{obs}$ is unchanged. For a candidate *p-value*, the all gene set will not be lower or equal to it unless the candidate *p-value* is itself 1. Thus $R_{obs}$ is unchanged. The effect on $R_{exp}$ is hard to determine, but for small empirical *p-values* should be proportionately small.

There are 98 VIP gene sets in [4], reduced to 53 when filtered for those containing at least 3 genes. For these and for the GO categories we used an FDR < 0.1 as a cut-off when discussing significant categories. There is only one SIV response genes set, so we only report the empirical *p-value* and treat *p-value* < 0.05 as significant. Note that this procedure does not allow the calculation of an FDR for the SIV set, nor over the family of tests (SIV gene set enrichment in all four subspecies) but we tested a strong *a* priori expectation that given the eastern PBSnj tail genes are enriched for viral immunity genes, this would be due to vervet SIV response genes. However, to estimate such an FDR, we used a resampling scheme: For each gene in the genome, we assign a weight, which is the proportion of SNPs in that gene compared to the genome as a whole. This is to correct for gene length bias. We make the intersect of all the SIV genes in each PBSnj tail. We then do weighted resampling from all genes in the genome to create sets of genes as large as the intersect set, and calculate an empirical *p-value* for each subspecies, as defined above. These empirical *p-values* are highly similar to those provided by GOWINDA, suggesting that our weighting scheme effectively controls for gene length bias.

We then calculate the FDR for each empirical *p-value*, with $R_{exp}$ summed over all four subspecies.

## Natural Host SIV responsive genes underpin the eastern PBSnj tail genic enrichment

We wanted to test if selection on natural host SIV responsive genes could be the reason that eastern chimpanzees exhibit the strongest signal of genetic adaptation. Our simple test is to hypothetically propose that if selection had not acted on the natural host SIV responsive genes then those genes would not contribute a SNP to the PBSnj eastern genic tail. Thus, we removed the genic tail SNPs from the 118 genes that are natural host SIV responsive and have SNPs in the outlier bin of the eastern PBS scores. However, we don't remove the genic SNPs within these genes that are in any of the other subspecies. This means we will affect the eastern genic enrichment, but not the enrichment of other subspecies. We argue that this answers the question "what would the eastern genic enrichment be if selection had not acted on these genes in eastern chimpanzees".

## Statistics

To test enrichment in *phastCons* scores and regulomeDB scores we use random resampling tests. For a candidate set of SNPs sized *n*, we randomly draw the same number of genic SNPs. For *phastCons* and regulomeDB we calculate the proportion of SNPs in a category.

For all resampling tests, *p-values* are estimated as 1 + *n* resamples > = observed (or < = observed as appropriate) / 1 + *n* resamples. Adding 1 to both the numerator and denominator ensures that resampling p-values do not equal 0, which is a downward biased estimate given finite resampling.

To test if the proportion of VIP category genes that were also vervet SIV responsive genes differed from null expectations, we used a binomial test of the proportion of VIP genes that were also SIV responsive compared to the null expectation, which is the proportion of all genes that are SIV responsive.

## Supporting information

**S1 Appendix. Signed differences in derived allele frequency ($\delta$) amongst human populations.**
(DOCX)

**S2 Appendix. Estimating the strength of background selection required to explain $\delta$ bin genic enrichments.**
(DOCX)

**S3 Appendix. Evidence for, and explanatory power of, differing strengths of BGS amongst chimpanzees.**
(DOCX)

**S4 Appendix. Population branch statistics.**
(DOCX)

**S5 Appendix. The relationship between divergence times and $N_e$ and the effects of BGS.**
(DOCX)

**S6 Appendix. Estimating the strength of background selection required to explain PBSnj tail genic enrichments in chimpanzees.**
(DOCX)

**S7 Appendix. Demography and the evidence of positive selection in central chimpanzees.**
(DOCX)

**S1 Fig. Genic enrichment in bins of signed difference in derived allele frequency ($\delta$), for human populations from the 1000 Genomes Phase III.** a, X-axis: $\delta$ is computed as the difference in derived allele frequency, for each pair of populations. Tail bins (the last bin in either end of $\delta$) contain those SNPs with the largest allele frequency differences. Numbers are of the genic SNPs in each tail bin. Y-axis: genic enrichment in each $\delta$ bin, computed as described in Methods. Shading represents the 95% CI (i.e. alpha = 0.05 for a two-tailed test) estimated by 200kb weighted block jackknife, b, The asymmetry of the genic enrichments in the $\delta$ tails is measured by taking their $\log_2$ ratio, thus 0 indicates a symmetric enrichment (equal enrichment in both $\delta$ tails). Dot = observed asymmetry, with size indicating the relative sample size (10, 20, 91 individuals). Horizontal lines represent confidence intervals estimated by 200kb weighted block jackknife (light = 95%, black = 99%, i.e. alpha = 0.05 or 0.01 for a two-tailed test).
(PDF)

**S2 Fig. Mean genic recombination rates across bins of signed difference in derived allele frequency ($\delta$).** X axes: Binned $\delta$, the difference in derived allele frequency, for each pair of populations. Y-axes: columns one and three: genic enrichment in each $\delta$ bin; columns two and four: mean genic recombination rate for each $\delta$ bin. Columns one and two: observed data. Columns three and four: analysis restricted to sites with recombination rate < 0.5 cM/Mb. Shading represents the 95% CI (i.e. alpha = 0.05 for a two-tailed test) estimated by 200kb weighted block jackknife. Light grey horizontal line represents: columns one and three expected genic enrichment; columns two and four, mean genic recombination rate.
(PDF)

**S3 Fig. Genic enrichment in bins of signed difference in derived allele frequency ($\delta$) compared to BGS simulations using genome average B. A,** X-axis: $\delta$ is computed as the difference in derived allele frequency, for each pair of chimpanzee subspecies. Tail bins (the last bin in either end of $\delta$) contain those SNPs with the largest allele frequency differences. Numbers are of the genic SNPs in each tail bin. Y-axis: genic enrichment in each $\delta$ bin (Methods). **B,** Genic enrichment eastern and central chimpanzee $\delta$, plotted separately due to a different Y-axis limit. NC = Nigeria-Cameroon. The asterisk shows significance of the asymmetry in the genic enrichment (* = 0.01). Shading represents the 95% CI (i.e. alpha = 0.05 for a two-tailed test) estimated by 200kb weighted block jackknife. Grey dashed lines represent simulations under increasing levels of background selection that best match different aspects of the data: lightest to darkest shades: B = 0.925 (excluding $\delta$ tail bins), 0.888 (all $\delta$ bins), and 0.863 (only $\delta$ tail bins).
(PDF)

**S4 Fig. The effect of background selection on patterns of neutral diversity in chimpanzees.**
**a**, Diversity levels at neutral sites as a function of the distance to the nearest gene. We calculated scale diversity (pi / divergence to macaque) in bins of distance to genic regions. We then rescaled scaled diversity for each subspecies so that the diversity was in the range 0–1. **b**, To further explore the effects of BGS on chimpanzee genomes we checked the correlation of density of functional sites with neutral diversity (pi). We used windows 500kb spaced at least 1MB

apart in the genome. Here, *rho* is the spearman rank partial correlation between windowed diversity and density of functional sites per window, controlling for recombination rate (the average rate per window). Each dot represents a bootstrap replicate (random sample of 500 kb windows). We calculated the partial rho for each bootstrap. Box plots show the median and interquartile ranges of the bootstrap replicates.
(PNG)

**S5 Fig. Stronger eastern BGS does not result in observed levels of $\delta$ tail bin genic enrichment asymmetry.** The asymmetry of the genic enrichments in the $\delta$ tails is measured by taking their $\log_2$ ratio, thus 0 indicates a symmetric enrichment (equal enrichment in both $\delta$ tails). We created coalescent simulations in which the strength of BGS was greater in eastern chimpanzees than other subspecies. For eastern chimpanzees we chose a fixed B = 0.825, as this B provided the best fit the eastern $\delta$ tail genic enrichment. All other subspecies had the same B, in the range of 0.900–0.850. A larger difference in B between subspecies results in a slight increase in asymmetry, but none of the simulated differences in BGS result in the observed asymmetry. Point = observed asymmetry. Horizontal lines represent confidence intervals estimated by 200kb weighted block jackknife (light = 95%, black = 99%, i.e. alpha = 0.05 or 0.01 for a two-tailed test). Grey vertical marks represent the $\delta$ tail asymmetry in simulations, under increasing levels of difference in background selection between eastern and other chimpanzees: lightest to darkest shades: All $B_{eastern}$ = 0.825; $B_{others}$ = 0.850, 0.863, 0.850, 0.888, 0.900.
(PDF)

**S6 Fig. Mean genic recombination rates across bins of PBSnj.** X axes: Binned *PBSnj*, for each subspecies. Y-axes: columns one and three: genic enrichment in each PBSnj bin; columns two and four: mean genic recombination rate for each PBSnj bin. Columns one and two: observed data. Columns three and four: analysis restricted to sites with recombination rate < 0.5 cM/Mb. Shading represents the 95% CI (i.e. alpha = 0.05 for a two-tailed test) estimated by 200kb weighted block jackknife. Light grey horizontal line represents: columns one and three expected genic enrichment; columns two and four, mean genic recombination rate.
(PDF)

**S7 Fig. Number of adaptive events in eastern chimpanzees.** a, Most 200kb blocks contain few PBSnj eastern outlier SNPs, but there is an extended right hand tail. b, we ranked blocks by the number of PBSnj eastern tail SNPs, then iteratively removed outlier genic SNPs. This results in a monotonically decreasing genic enrichment, and the removal of eight blocks is required to reduce the genic enrichment of the PBSnj eastern tail to overlap the 95% CI of the PBSnj central tail, and 19 blocks to reduce it below the level of the point estimate of the central PBSnj tail. We could alternatively order the windows by the total number of outlier SNPs i.e. without regard to genic vs. non-genic. Doing so increases our estimated range of sweeps to 15–26. But we note that the genic enrichment does monotonically decrease with block removal (c). This is partly due to the arbitrary nature of the definition of genic, as it implies that there are some 200 kb blocks that have more non-genic than genic outlier SNPs contained within them, and this may very well change if the definition of genic was changed from transcription start and end sites +- 2kb. (d) Lastly, we randomly shuffled the removal order of the 200-kb blocks. We did so for 1000 random shuffles of the block order (A single random shuffle is shown). We find that the median number of blocks (i.e. sweeps) across random shuffles is 165 to match the upper 95% CI of the central chimpanzee estimate (middle 90% quantile range 114–221; min = 78, max = 278) increased to 273 to match the central chimpanzee point estimate of genic enrichment (middle 90% quantile range 214–329; min = 162, max = 381). Such a procedure is likely an overestimate, as most of the removal steps are those removing 1 to 9

genic outlier SNPs (panel a), resulting in minimal reduction of the genic enrichment.
(PDF)

**S8 Fig. Number of sweeps in the chromosome 3 chemokine receptor cluster of central chimpanzees.** X axis: position along chromosome 3 (Mb). Plotted in the upper panel are the PBSnj central scores in the region encompassing *CCR3*, *CCR9*, and *CXCR6*. An independent cluster of high PBSnj scores is associated with each candidate gene. Each point represents one PBSnj score, colour has an alpha = 30% to reduce over plotting. Haplotypes are plotted in the central panel. Yellow ticks are derived alleles, blue are ancestral, while white is space so that each tick aligns with PBSnj scores. Inspection indicates that there is a degree of haplotype scrambling between each of the candidate genes. Lastly, we depict the genes in this region in the lower panel.
(PNG)

**S9 Fig. Observed and Simulated Site Frequency Spectra.** We plot the Site Frequency Spectrum (SFS) for each chimpanzee subspecies. X axes: derived allele count. Y axes: proportion. Black: observed. Green: simulated. Simulated counts come from 25 million 2kb loci simulated with msms, using the chimpanzee demography specified in Methods.
(PDF)

**S10 Fig. The genome wide distribution of B values.** Violin plots are used to visualise the distribution of B values for non-genic and genic sites from McVicker, designated non-genic an "0.88" respectively. B = 0.88 is the average B ratio of genic / non-genic B values. We added fixed constants to all genic B values to modify the genome average B, in 0.01 steps from 0.99–0.93.
(PDF)

**S11 Fig. Deriving the PBSnj statistic. a,** PBS is just a simple arithmetic function of pairwise $F_{ST}$ values for a group of three taxa or populations. **b,** The configuration or choice of populations determines the information content of PBS. In each panel are the spearman's rho correlations between different PBS configurations, and between PBS and our new statistic PBSnj for a simple four population model (described in Appendix 4). In each case Pop A is the focal population. $PBS_{ABC}$ and $PBS_{ABD}$ are highly correlated but not identical indicating that incorporating both Pops C and D would refine the identification of Pop A specific differentiated variants. PBSnjA, which utilises information from all four populations is more highly correlated with both $PBS_{ABC}$ and $PBS_{ABD}$ than they are with each other. Alpha = 10% for plotted points to reduce over saturation. **c,** For each statistic, we plot the site frequency spectrum (SFS) for each of the four populations for sites identified as outliers in Pop A. PBSnj clearly finds those sites differentiated in Pop A, and better than either $PBS_{ABC}$ and $PBS_{ABD}$. In the standard PBS, the SFS in the species not included in the PBS configuration has a more uniform distribution, indicating that some sites identified as PBS frequency outliers in Pop A are not true population specific outliers.
(PNG)

**S12 Fig. Effect of reduced Ne on PBSnj genic enrichments.** In a simple four population model, we modelled genic regions as those with a B = 0.9. In population 2, we simulated four effective population size ratios (1, 0.9, 0.5, 0.1). $N_e$ ratios of 0.5 and 0.1 result in a reduced genic enrichment given the same strength of background selection. X-axes: PBS scaled to take values in the range 0–1, per subspecies. Y-axes: Genic enrichment.
(PDF)

**S13 Fig. Scaled PBSnj bin genic enrichment for all chimpanzee subspecies.** PBS scaled to take values in the range 0–1. Y-axes: Genic enrichment computed as described in Fig 2. Shading represents the 95% CI (i.e. alpha = 0.05 for a two-tailed test) estimated by 200kb weighted block jackknife. B: $\log_2$ ratio of the eastern versus western, Nigeria-Cameroon and central PBSnj tails (PBS $>$ = 0.8) genic enrichment. A,B Grey dashed (A) or vertical (B) lines represent the PBSnj genic enrichment in simulations, under increasing levels of background selection that best match different aspects of $\delta$, as described in Figs 2 and 3: lightest to darkest shades: B = 0.93 (excluding $\delta$ tail bins), 0.92 (all $\delta$ bins), and 0.88 (unmodified genic B values form McVicker).
(PDF)

**S1 Table. Signed difference in derived allele frequency, genic and non-genic tail counts.**
(XLSX)

**S2 Table. Observed and simulated $\delta$ bin genic enrichments.**
(XLSX)

**S3 Table. Observed and model chimpanzee subspecies $F_{ST}$.**
(XLSX)

**S4 Table. Fit of simulated to observed genic enrichments across $\delta$ bins.**
(XLSX)

**S5 Table. $\log_2$ ratio of eastern and central chimpanzee $\delta$ tail bin genic enrichments with different strengths of background selection.**
(XLSX)

**S6 Table. Model-based reduction of neutral diversity in chimpanzee subspecies. Models are tested for their ability to explain diversity as a function of distance to functional sites.**
(XLSX)

**S7 Table. Effect of divergence on $\delta$ tail SNP number.**
(XLSX)

**S8 Table. Effect of $N_e$ on $\delta$ tail SNP number.**
(XLSX)

**S9 Table. Fitting BGS to match observed PBSnj tail genic enrichments.**
(XLSX)

**S10 Table. Non-synonymous PBSnj eastern tail SNPs.**
(XLSX)

**S11 Table. PBSnj tail SNP regulomeDB enrichments.**
(XLSX)

**S12 Table. PBSnj tail SNP conservation/phastCons score enrichments.**
(XLSX)

**S13 Table. Effect of Ancestral Allele estimation on eastern vs. central chimpanzee $\delta$ bin genic enrichments.**
(XLSX)

**S14 Table. PBSnj Eastern GO enrichment.**
(XLSX)

**S15 Table. PBSnj Central GO enrichment.**
(XLSX)

**S16 Table. PBSnj Nigeria-Cameroon GO enrichment.**
(XLSX)

**S17 Table. PBSnj Western GO enrichment.**
(XLSX)

**S18 Table. PBSnj Eastern VIP enrichment.**
(XLSX)

**S19 Table. PBSnj Central VIP enrichment.**
(XLSX)

**S20 Table. PBSnj Nigeria-Cameroon VIP enrichment.**
(XLSX)

**S21 Table. PBSnj Western VIP enrichment.**
(XLSX)

**S22 Table. SIV responsive gene enrichment tests.**
(XLSX)

**S23 Table. PBSnj eastern tail and SIV responsive genes SNP regulomeDB enrichment.**
(XLSX)

**S24 Table. PBSnj Eastern and SIV gene GO enrichment.**
(XLSX)

**S25 Table. PBSnj Eastern SIV co-expression module enrichment.**
(XLSX)

## Acknowledgments

We thank: Fabrizio Mafessoni, Linda Vigilant, Mimi Arandjelovic, Paolo Gratton, Hjalmar Kühl and Lauren White of the Max Planck Institute for Evolutionary Anthropology for helpful discussions and/or comments on the manuscript; Alan Boyle for providing regulomeDB scores; Hannes Svardal for extensive discussions and comments on the manuscript; the anonymous reviewers whose comments and suggestions on this and previous versions of this manuscript, greatly improved the current manuscript.

## Author Contributions

**Conceptualization:** Joshua M. Schmidt, Sergi Castellano, Aida M. Andrés.

**Data curation:** Joshua M. Schmidt, Marc de Manuel, Tomas Marques-Bonet.

**Formal analysis:** Joshua M. Schmidt.

**Funding acquisition:** Aida M. Andrés.

**Investigation:** Joshua M. Schmidt, Aida M. Andrés.

**Methodology:** Joshua M. Schmidt.

**Project administration:** Joshua M. Schmidt, Sergi Castellano, Aida M. Andrés.

**Resources:** Sergi Castellano, Aida M. Andrés.

**Supervision:** Sergi Castellano, Aida M. Andrés.

**Visualization:** Joshua M. Schmidt.

**Writing – original draft:** Joshua M. Schmidt, Sergi Castellano, Aida M. Andrés.

**Writing – review & editing:** Joshua M. Schmidt, Marc de Manuel, Tomas Marques-Bonet, Sergi Castellano, Aida M. Andrés.

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
