## [Decision Letter · Decision Letter 0]

12 Aug 2019

Dear Dr Schmidt,

Thank you very much for submitting your Research Article entitled 'The impact of genetic adaptation on chimpanzee subspecies differentiation' to PLOS Genetics. Your manuscript was fully evaluated at the editorial level and by independent peer reviewers. The reviewers appreciated the attention to an important problem, but raised some substantial concerns about the current manuscript. Based on the reviews, we will not be able to accept this version of the manuscript, but we would be willing to review again a much-revised version. We cannot, of course, promise publication at that time.

If you decide to revise the manuscript for further consideration at PLOS Genetics, please aim to resubmit within the next 60 days, unless it will take extra time to address the concerns of the reviewers, in which case we would appreciate an expected resubmission date by email to plosgenetics@plos.org.

[LINK]

We are sorry that we cannot be more positive about your manuscript at this stage. Please do not hesitate to contact us if you have any concerns or questions.

Yours sincerely,

Takashi Gojobori

Associate Editor

PLOS Genetics

Bret Payseur

Section Editor: Evolution

PLOS Genetics

Reviewer's Responses to Questions

**Comments to the Authors:**

Reviewer #1: I think this manuscript is an excellent contribution to the literature focusing on evolutionary responses to environmental challenges in a species closely related to ours.

The authors have demonstrated an ability to go beyond the limitations of previous studies of adaptation in chimpanzees to identify the targets of selection (low sample sizes). I particularly liked the attempt to identify the unique genetic adaptations of each subspecies. The authors found evidence for local adaptation, which, as they point out, contrasts with results from comparable literature on humans. As such I am confident that the article will gather readers' attention, and generate valuable discussion.

I found the methodology adequate, and appreciated the care with which the authors drew biological conclusions from their data set. I am happy to recommend publication of this article.

Reviewer #2: In their manuscript, the authors provide overall convincing evidence for a potential role of SIV in adaptive signals in specific chimpanzee populations. Overall I liked the manuscript, and the contribution made to the growing evidence that viruses are probably the strongest selective pressure shaping host genome variation. My main issue is that multiple factors could have decreased the power of the analysis, that need to be more extensively discussed not only at the end of the manuscript in the discussion, but in the introduction as a priori known limitations of the methods used:

1) Differentiation only works when the selective pressures and selection signals are not shared. There could be a lot of unseen shared positive selection going on that is not detected by the differentiation/binning strategy. It is especially relevant in the case of selection against viruses, as viruses have the remarkable property of being a traveling selective pressure that tends to not be geographically bound as other pressures such as climate, or available food.

2) The authors mention it late in the discussion, but increased drift in smaller chimp populations could mask an existing genome-wide selection signal, which does not imply that there is no selection but that the authors would have less power to detect it. This needs to be discussed right form the introduction and when results are presented, as opposed to just saying there is only evidence of strong selection in Eastern chimps in these parts of the manuscript. It is the old issue of the outliers approach, that absence of evidence is not evidence of absence.

3) If selection in genic regions was strong, it could pervade quite far in non-genic regions, which will further reduce the power of the genic vs. non-genic strategy. This also needs to be discussed, together with the fact that there may actually be also positive selection in non-genic regions, in non-genic regulatory elements.

4) P4L80: Limited role of hard sweeps. The authors cannot possibly ignore the work of Enard, Messer and Petrov, and Schrider and Kern. Enard et al. showed that genome-wide sweeps signals in humans are masked by background selection, and used far more rigorous controls than previous studies to do so (Genome Research 2014). Schrider and Kern used more powerful machine learning approaches to find many hard and soft sweeps signals in the human genome, with very strong enrichments at genes that interact with viruses (MBE 2017). The authors need to be much more diligent and cite the most up-to-date evidence, especially evidence based on more rigorous controls and more statistical power compared to the older papers they cite (papers #10 and #23). The view provided in the introduction is as of now quite biased. As always, absence of evidence from underpowered studies is not surprising, and certainly not evidence of absence.

P6L123: Lower recombination at genic sites compared to non-genic could also increase differentiation without natural selection. The authors need to control for that, for example by comparing genic sites with non-genic sites with similar recombination levels, or show that either recombination rates between genic and non-genic sites do not differ significantly in the first place, or recombination is higher at genic sites. The question is important because past studies such as Myers et al. Science 2005 (https://science.sciencemag.org/content/310/5746/321.long) have shown quite drastic variation of recombination rates as a function of the distance to the nearest gene. Figure 3 in Myers et al. shows that non-genic sites far from genes may have strongly decreased recombination rates, which may have decreased the power of the present analysis. Conversely, according to the same figure non-genic regions right outside of genic ones have elevated recombination rates. So, depending on the distance of non-genic control sites, they could either be overall too conservative (lower overall recombination than genic), or conversely too liberal (higher overall recombination than genic).

P6L129: “not relying on the patterns of linked variation”. Nearby SNPs are linked to each other, and physically clustered. The binning strategy does not erase the fact that SNPs are clustered together. The claim that the strategy does not rely on linked variation does not mean it is immune to it, and therefore needs to be removed. Linked non-genic SNPs close enough to genic ones may well be influenced by positive selection through linkage, which could have greatly decreased the power of the genic vs. non-genic strategy. This is something that the authors should mention and discuss, especially in the context of strong positive selection that can create extended differentiation signals.

P13L306: Please indicate if the NS site enrichment could be due to the fact that the differentiated loci happen to have more coding sequences compared to the genic background, as suggested by the similar NS/S ratios.

P15L357. Table 1. It is intriguing that the VIPs from HIV are not significant given that HIV likely shares the most interactions with SIV. The authors should discuss the lack of signal at VIPs for HIV.

P19L453: Again, the most relevant paper to cite on human soft sweeps is Schrider and Kern, MBE 2017, given that the authors actually show abundant soft sweeps from standing variation in human genomes. I am honestly quite surprised that the authors seem to ignore this paper.

**Have all data underlying the figures and results presented in the manuscript been provided?**

Reviewer #1: Yes

Reviewer #2: Yes

PLOS authors have the option to publish the peer review history of their article (what does this mean?). If published, this will include your full peer review and any attached files.

Reviewer #1: No

Reviewer #2: No

---

## [Decision Letter · Decision Letter 1]

17 Oct 2019

Dear Dr Schmidt,

We are pleased to inform you that your manuscript entitled "The impact of genetic adaptation on chimpanzee subspecies differentiation" has been editorially accepted for publication in PLOS Genetics. Congratulations!

Yours sincerely,

Takashi Gojobori

Associate Editor

PLOS Genetics

Bret Payseur

Section Editor: Evolution

PLOS Genetics

Comments from the reviewers (if applicable):

Reviewer's Responses to Questions

Comments to the Authors:

Please note here if the review is uploaded as an attachment.

Reviewer #2: The authors have thoroughly addressed my previous concerns. The new controls for recombination heterogeneity were particularly important and now make the results stronger.

Have all data underlying the figures and results presented in the manuscript been provided?

Large-scale datasets should be made available via a public repository as described in the 

PLOS Genetics

data availability policy, and numerical data that underlies graphs or summary statistics should be provided in spreadsheet form as supporting information.

Reviewer #2: Yes

PLOS authors have the option to publish the peer review history of their article (what does this mean?). If published, this will include your full peer review and any attached files.

Do you want your identity to be public for this peer review?

 For information about this choice, including consent withdrawal, please see our Privacy Policy.

Reviewer #2: No

**Data Deposition**

http://datadryad.org/submit?journalID=pgenetics&manu=PGENETICS-D-19-01140R1

Press Queries

---

## [Editor Report · Acceptance letter]

18 Nov 2019

PGENETICS-D-19-01140R1 

The impact of genetic adaptation on chimpanzee subspecies differentiation 

Dear Dr Schmidt, 

We are pleased to inform you that your manuscript entitled "The impact of genetic adaptation on chimpanzee subspecies differentiation" has been formally accepted for publication in PLOS Genetics! Your manuscript is now with our production department and you will be notified of the publication date in due course.

With kind regards,

Matt Lyles

PLOS Genetics

On behalf of:
